# Core PCP mutations affect short-time mechanical properties but not tissue morphogenesis in the *Drosophila* pupal wing

Romina Piscitello-Gómez[1,2†], Franz S Gruber[1,3†], Abhijeet Krishna[1,2,4], Charlie Duclut[5,6,7], Carl D Modes[1,2,4], Marko Popović[2,4,6], Frank Jülicher[2,4,6], Natalie A Dye[1,2,8]*, Suzanne Eaton[1,2,4,9‡]

[1]Max Planck Institute of Molecular Cell Biology and Genetics, Dresden, Germany; [2]DFG Excellence Cluster Physics of Life, Technische Universität Dresden, Dresden, Germany; [3]National Phenotypic Screening Centre, University of Dundee, Dundee, United Kingdom; [4]Center for Systems Biology Dresden, Dresden, Germany; [5]Laboratoire Physico-Chimie Curie, CNRS UMR 168, Institut Curie, Université PSL, Sorbonne Université, Paris, France; [6]Max Planck Institute for Physics of Complex Systems, Dresden, Germany; [7]Université Paris Cité, Laboratoire Matière et Systèmes Complexes, Paris, France; [8]Mildred Scheel Nachwuchszentrum P2, Medical Faculty, Technische Universität Dresden, Dresden, Germany; [9]Biotechnologisches Zentrum, Technische Universität Dresden, Dresden, Germany

*For correspondence:
natalie_anne.dye@tu-dresden.de

†These authors contributed equally to this work

‡Deceased

Competing interest: The authors declare that no competing interests exist.

**Abstract** How morphogenetic movements are robustly coordinated in space and time is a fundamental open question in biology. We study this question using the wing of *Drosophila melanogaster*, an epithelial tissue that undergoes large-scale tissue flows during pupal stages. Previously, we showed that pupal wing morphogenesis involves both cellular behaviors that allow relaxation of mechanical tissue stress, as well as cellular behaviors that appear to be actively patterned (Etournay et al., 2015). Here, we show that these active cellular behaviors are not guided by the core planar cell polarity (PCP) pathway, a conserved signaling system that guides tissue development in many other contexts. We find no significant phenotype on the cellular dynamics underlying pupal morphogenesis in mutants of core PCP. Furthermore, using laser ablation experiments, coupled with a rheological model to describe the dynamics of the response to laser ablation, we conclude that while core PCP mutations affect the fast timescale response to laser ablation they do not significantly affect overall tissue mechanics. In conclusion, our work shows that cellular dynamics and tissue shape changes during *Drosophila* pupal wing morphogenesis do not require core PCP as an orientational guiding cue.

## Editor's evaluation

This valuable study shows that the core PCP pathway, which commonly orients morphogenetic processes in development, does not establish global cues for cellular movements in the *Drosophila* pupal wing. Using a combination of laser-ablation experiments and mathematical modelling, it provides compelling evidence that the core PCP pathway only affects the fast timescale cellular response, but does not appear to drive overall tissue dynamics. While the signalling pathway guiding this morphogenetic process remains to be elucidated, these relevant findings challenge the role of the core PCP pathway in morphogenesis.

## Introduction

The spatial–temporal pattern of mechanical deformation during tissue morphogenesis is often guided by patterns of chemical signaling. Precisely how chemical signaling couples with the mechanics of morphogenesis, however, remains an active area of research. One conserved chemical signaling pathway that is known to be patterned across tissues is the core planar cell polarity (PCP) pathway, composed of a dynamic set of interacting membrane proteins that polarizes intracellularly within the plane of a tissue. Tissue-scale alignment of this pathway is known to orient cellular structures, such as hairs and cilia, and influence dynamic cellular behaviors during morphogenesis, such as cellular movements and cell divisions, through interactions with the cytoskeleton (reviewed in *Devenport, 2014*; *Butler and Wallingford, 2017*; *Deans, 2021*).

Here, we examine a potential role for the core PCP pathway in the dynamics and mechanics of morphogenesis using the *Drosophila* pupal wing. The *Drosophila* wing is a flat epithelium that can be imaged at high spatial–temporal resolution *in vivo* during large-scale tissue flows that elongate the wing blade (*Aigouy et al., 2010*; *Etournay et al., 2015*; *Guirao et al., 2015*). During the pupal stage, the proximal hinge region of the wing contracts and pulls on the blade region, generating mechanical stress that is counteracted by marginal connections mediated by the extracellular matrix protein Dumpy (*Etournay et al., 2015*; *Ray et al., 2015*). As a consequence, the tissue elongates along the proximal–distal (PD) axis and narrows along the anterior–posterior (AP) axis to resemble the adult wing. Both cell elongation changes and cell rearrangements are important for tissue deformation. To some extent, mechanical stress induces these cell behaviors. However, the reduction of mechanical stress in a *dumpy* mutant does not completely eliminate cell rearrangements, suggesting that there could be other patterning cues that drive oriented cell rearrangements (*Etournay et al., 2015*). We therefore wondered whether chemical PCP systems could orient cell behaviors, such as cell rearrangements, during pupal blade elongation flows.

In the *Drosophila* wing, there are two PCP systems termed Fat and core PCP (*Matis and Axelrod, 2013*; *Adler, 2012*; *Devenport, 2014*; *Butler and Wallingford, 2017*). The Fat PCP system consists of two cadherins Fat and Dachsous, a cytoplasmic kinase Four-jointed, and an atypical myosin Dachs. The core PCP system is composed of two transmembrane proteins Frizzled (Fz) and Flamingo or Starry night (Fmi, Stan), the transmembrane protein Strabismus or Van Gogh (Stbm, Vang), and the cytosolic components Dishevelled (Dsh), Prickle (Pk), and Diego (Dgo).

Our group has shown that tissue-scale patterns of PCP emerge during larval stages and then are dynamically reoriented during pupal tissue flows (*Sagner et al., 2012*; *Aigouy et al., 2010*; *Merkel et al., 2014*). At the onset of blade elongation flows, both systems are margin oriented, however as morphogenesis proceeds, core PCP reorients to point along the PD axis, whereas Fat PCP remains margin oriented until very late, when it reorients toward veins (*Figure 1—figure supplement 1A, B*; *Merkel et al., 2014*). Whether these PCP systems and their reorientation influence tissue dynamics and mechanics during blade elongation flows is unknown.

The core PCP pathway has been shown to influence numerous processes in *Drosophila* tissue development. These include hexagonal cell packing in the late pupal wing (*Classen et al., 2005*; *Sugimura et al., 2016*), as well as patterning of ommatidial clusters in the developing eye (*Zheng et al., 1995*; *Jenny, 2010*), orientation of cell division in sensory organ precursors (*Gho and Schweisguth, 1998*), formation of joints in the legs (*Capilla et al., 2012*), and regulation of tracheal tube length (*Chung et al., 2009*). In many cases, the mechanism connecting the core PCP pathway to cell dynamics and tissue mechanics is unclear. Recent studies suggest, however, that core PCP may act in concert with Nemo kinase to regulate cell rearrangements in the eye (*Mirkovic et al., 2011*; *Founounou et al., 2021*) and with the *Drosophila* NuMA ortholog Mud to orient cell division orientation in the sensory organ precursors (*Ségalen et al., 2010*).

Here, we examine cellular dynamics in tissues mutant for core PCP and we find that they are largely unperturbed, indicating that core PCP does not have an essential role in organizing global patterns of cell rearrangements in the pupal wing. We also performed an extensive analysis of the mechanics using laser ablation, developing a rheological model to interpret the results. We find that mutants in core PCP differ from wild type in the initial recoil velocity upon laser ablation. We find, however, that this difference is produced from the very fast timescale response, which does not appear to affect morphogenesis and overall tissue stresses, consistent with the lack of phenotype in cellular dynamics.

## Results

### Core PCP does not guide cellular dynamics during pupal blade elongation flows

To investigate the role of core PCP in orienting cell behaviors during pupal blade elongation flows, we analyzed cell dynamics in wild type (*wt*) and three different *core PCP* mutant tissues: *prickle* (*pk³⁰*, abbreviated as *pk*), *strabismus* (*stbm⁶*, abbreviated as *stbm*), and *flamingo* (*fmi^frz3*, aka *stan^frz3*, abbreviated as *fmi*). In *pk*, the core and Fat PCP systems remain aligned together toward the margin and the magnitude of Stbm polarity is reduced (*Merkel et al., 2014*). The mutants *stbm* and *fmi* are strong hypomorphs, where the core PCP network is strongly reduced (*Figure 1—figure supplement 1B*; *Merkel et al., 2014*). We analyzed shape changes of the wing blade during blade elongation flows and decomposed these changes into contributions from cell elongation changes and cell rearrangements, which include cell neighbor exchanges, cell divisions, cell extrusions, and correlation effects (*Figure 1*; *Etournay et al., 2015*; *Merkel et al., 2017*).

In *wt*, the wing blade elongates along the PD axis (blue line in *Figure 1D*). Cells first elongate along the PD axis and then relax to more isotropic shapes (green line in *Figure 1D*). Cell rearrangements, however, go the opposite direction, initially contributing to AP deformation, before turning around to contribute to PD deformation (magenta line in *Figure 1D*). We introduce here a relative timescale, where we measure time in hours relative to the peak in cell elongation (hRPCE). This new scale allows us to handle variation in the timing of the onset of the blade elongation flows, which we have observed recently (see Appendix 1).

In *core PCP* mutants, we find that the dynamics of tissue shear, cell elongation changes, and cell rearrangements, when averaged across the entire blade, occur normally (*Figure 1D–D''*). We observe that by the end of the process, only slightly lower total shear appears to occur in the *core PCP* mutants, caused by slightly less cell rearrangements, but these subtle changes are not statistically significant (*Figure 1—figure supplement 2C*). The cellular dynamics contributing to isotropic tissue deformation are also broadly the same between *wt* and *core PCP* mutant tissues (*Figure 1—figure supplement 2D, E*). We also looked for differences in the behavior of regions of the wing blade subdivided along the PD axis (*Figure 1—figure supplement 3*), as previous work has shown that distal regions of the wing blade shear more at early times, whereas proximal regions start deforming later (*Merkel et al., 2017*). Again, we do not find strong differences between *core PCP* mutants and *wt* when we subdivide the wing into regions along the PD axis (*Figure 1—figure supplement 3*, *Figure 1—figure supplement 4*).

From this analysis, we conclude that core PCP is not required to determine the global patterns of cell dynamics during blade elongation flows. Interestingly, *core PCP* mutants do have a subtle but significant phenotype in the adult wing shape: *pk* and *stbm* (but not *fmi*) mutant wings are slightly rounder and wider than *wt* (*Figure 1—figure supplement 5I*). In principle, these small differences could arise after the blade elongation flows studied here. However, it is also possible that the we could not reliably detect these subtle differences in pupal wings due to the small number of wings per genotype that we were able to analyze (*n* = 2−4). To illustrate this point, we used the pool of adult wings (*n* = 53 for *wt*, *n* = 47 for *pk*, *n* = 74 for *stbm*, and *n* = 56 for *fmi*), where the phenotype is significant, to understand the probability that a sample of smaller size *m* would provide a significant signal, see *Figure 1—figure supplement 5J*. For *m* = 3, corresponding to the number of pupal wings we analyzed, we find that only about 20% of samples show a significant phenotype. In other words, if the same magnitude of difference occurred during the blade elongation flows as in the adult, we would have only about 20% chance to observe it. Therefore, core PCP could subtly influence the cell dynamics occurring at this stage. To investigate this possibility, we next looked for a possible difference in mechanical stresses in *core PCP* mutants.

### A rheological model for the response to laser ablation

We investigated cell and tissue mechanics in *core PCP* mutants using laser ablation in a small region of the wing blade. We used a region located between the second and third sensory organs in the intervein region between the L3 and L4 longitudinal veins, which is a region that is easy to identify throughout blade elongation flows (*Figure 2A*). We cut three to four cells in a line along the AP axis and measured the displacement of the tissue (*Figure 2A*, *Video 1*). We calculate the initial recoil velocity *v* by measuring the average displacement of ablated cell membranes at the first observed

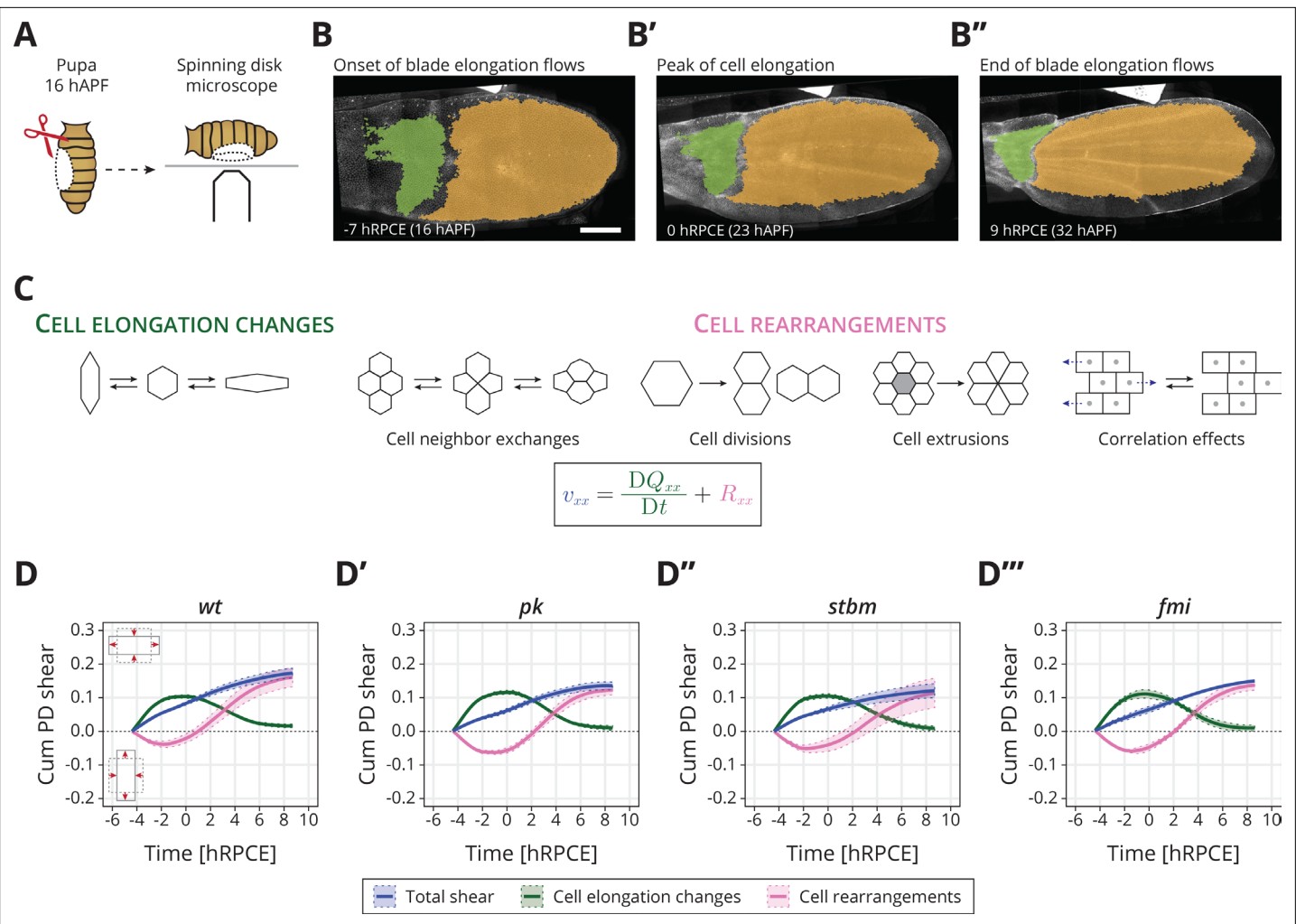

**Figure 1.** Core planar cell polarity (PCP) does not orient cellular behaviors and tissue reshaping during pupal blade elongation flows: (**A**) Cartoon of pupal wing dissection at 16 hAPF and imaging using a spinning disk microscope. (**B–B″**) Images of a *wt* wing at −7, 0, and 9 hRPCE (for this movie these times correspond to 16, 23, and 32 hAPF). The green and orange regions correspond to the hinge and blade, respectively. Anterior is up; proximal to the left. Scale bar, 100 μm. (**C**) Schematic of the cellular contributions underlying anisotropic tissue deformation. The tissue shear rate component $v_{xx}$, which quantifies the rate of anisotropic tissue deformation along the proximal–distal wing axis, is decomposed into deformations arising from the rate of change of cell shapes $\mathrm{D}Q_{xx}/\mathrm{D}t$ and the deformations arising from the cellular rearrangements $R_{xx}$ (*Etournay et al., 2015*; *Merkel et al., 2017*). Total shear is the sum of cell elongation changes (green) and cell rearrangements (magenta). (**D–D‴**) Accumulated proximal–distal (Cum PD) tissue shear during blade elongation flows in the blade region averaged for (**D**) *wt* (n = 4), (**D′**) *pk* (n = 3), (**D″**) *stbm* (n = 3), and (**D‴**) *fmi* (n = 2) movies. Solid line indicates the mean, and the shaded regions enclose ± standard error of the mean (SEM). Differences in total accumulated shear are not statistically significant (*Figure 1—figure supplement 2C*). Time is relative to peak cell elongation (hRPCE).

The online version of this article includes the following source data and figure supplement(s) for figure 1:

**Source data 1.** Numerical data for *Figure 1D–D‴*, accumulated proximal–distal tissue shear during blade elongation flows in the blade region for *wt* and *core PCP* mutants.

**Figure supplement 1.** Reorganization of the core and Fat planar cell polarity (PCP) systems during pupal blade elongation flows.

**Figure supplement 2.** Quantification of final pupal tissue deformation and cellular contributions to isotropic tissue area.

**Figure supplement 2—source data 1.** Numerical data of *Figure 1—figure supplement 2C*.

**Figure supplement 2—source data 2.** Numerical data of *Figure 1—figure supplement 2D*.

**Figure supplement 2—source data 3.** Numerical data of *Figure 1—figure supplement 2E*.

**Figure supplement 3.** Regional analysis of tissue shear in the hinge and four blade subregions.

**Figure supplement 3—source data 1.** Numerical data of *Figure 1—figure supplement 3G*.

**Figure supplement 4.** Statistics of final shear in the blade subregions and its cellular contributions.

*Figure 1 continued on next page*

*Figure 1 continued*

**Figure supplement 4—source data 1.** Numerical data of *Figure 1—figure supplement 4H*.

**Figure supplement 5.** Adult wing shape quantification and random sampling.

**Figure supplement 5—source data 1.** Numerical data of *Figure 1—figure supplement 5I*.

**Figure supplement 5—source data 2.** Numerical data of *Figure 1—figure supplement 5J*.

timepoint after the ablation, $\delta t = 0.65s$ (see Materials and methods, Linear laser ablations to calculate the initial recoil velocity). Previously, we reported that initial recoil velocity measured along the PD axis in *wt* peaks around −8 hRPCE (20 hAPF in *Iyer et al., 2019*), and therefore we first focus on this timepoint. We find that *core PCP* mutants have significantly lower initial recoil velocity (*Figure 2B*, *Figure 2—figure supplement 1A*), suggesting that there is a mechanical defect in these mutants.

As initial recoil velocity is often used as a proxy for mechanical stress (e.g. *Mayer et al., 2010*; *Etournay et al., 2015*; *Iyer et al., 2019*; *Farhadifar et al., 2007*), this result seems to suggest that the PCP mutant wings generate less mechanical stress during blade elongation flows, even though the cellular dynamics are at best only subtly perturbed. To explore this phenotype in more detail, we considered that the response to laser ablation is not exactly a direct measure of mechanical stress, as it is also affected by cellular material properties. We thus further analyzed the full kinetics of the linear laser ablations, focusing on the *pk* mutant, and developed a rheological model to interpret the results. When plotting displacement of the nearest bond to ablation over time, we realized that a single exponential relaxation cannot account for the observed behavior (*Figure 2—figure supplement 1B*, *right*). We obtained a good fit of the data by introducing a second relaxation timescale (*Figure 2C*). The slow timescale (~20 s) accounts for most of the timecourse of displacement changes, but the fast timescale (<1 s) is required to account for first 5–10 datapoints, see *Figure 2—figure supplement 1B*, *left*. We therefore developed a model consisting of two Kelvin–Voigt (KV) elements in series (*Figure 2D*) to represent the tissue after ablation. The two KV elements have different elastic constants ($k_f$ and $k_s$) and viscosities ($\eta_f$ and $\eta_s$). Before ablation, the system is subjected to a constant stress ($\sigma$) and contains a spring with elastic constant $k$, which represents the cell patch that will be ablated. Upon ablation, the third spring is removed which leads to change in strain of our rheological model. We represent this strain by a displacement $\Delta x$ as a function of time given by

$$\Delta x(t) = X_f(1 - e^{-t/\tau_f}) + X_s(1 - e^{-t/\tau_s}), \tag{1}$$

where $X_f = \sigma\kappa/k_f$ is the displacement associated with the fast timescale, $\tau_f = \eta_f/k_f$, and $X_s = \sigma\kappa/k_s$ is the displacement associated with the slow timescale, $\tau_s = \eta_s/k_s$. Here, $\kappa = k/(k + \bar{k})$ is the fraction of the overall system elasticity lost due to ablation (see Materials and methods, Kymograph analysis and fit to model) and $\bar{k} = k_s k_f/(k_s + k_f)$ is the elasticity of the two KV elements connected in series. With this model, we presume the properties of the ablated cell itself, including its membrane, adhesion proteins, and acto-myosin cortex likely dominate the fast timescale response. The slow timescale response is a collective effect emerging from the ablated cell together with its surrounding cellular network.

We analyzed the experimentally measured displacement over time for each ablation and then fit the data to our model with four parameters ($X_f$, $X_s$, $\tau_f$, and $\tau_s$) (*Equation 1*, *Figure 2E–E'''*). Surprisingly, we find that the only parameter that changes between *pk* and *wt* is $X_f$, the displacement associated with the fast timescale (*Figure 2E–E'''*). To interpret this result, we consider that these four fitted parameters constrain the five mechanical model parameters (*Figure 2D*) but do not provide a unique solution. Since only one measured parameter changes, we asked what is the simplest set of model parameter changes that could have such an effect. To this end, we first note that the measured values of $X_f$ and $X_s$ ($1.8 - 2.6 \, \mu m$ vs $6 - 8 \, \mu m$, respectively) indicate $k_f \gg k_s$ and therefore the overall elasticity of our rheological model is largely determined by the elasticity of the slow relaxation $\bar{k} \approx k_s$. If we also consider that the contribution to the elasticity of the cellular patch from the ablated cells, represented by $k$ in the model, is small, then we can approximate $\kappa \approx k/k_s$ and therefore $X_f \approx \sigma k/(k_s k_f)$ (see Materials and methods, Kymograph analysis and fit to model). Is the observed phenotype in the fast timescale displacement $X_f$ due to a change in tissue stress $\sigma$ or a change in the elastic constants?

To address this question, we sought to probe mechanical stress in the *wt* and *pk* mutant, independent of the ablation recoil velocity. To do so, we used a method called ESCA (elliptical shape after

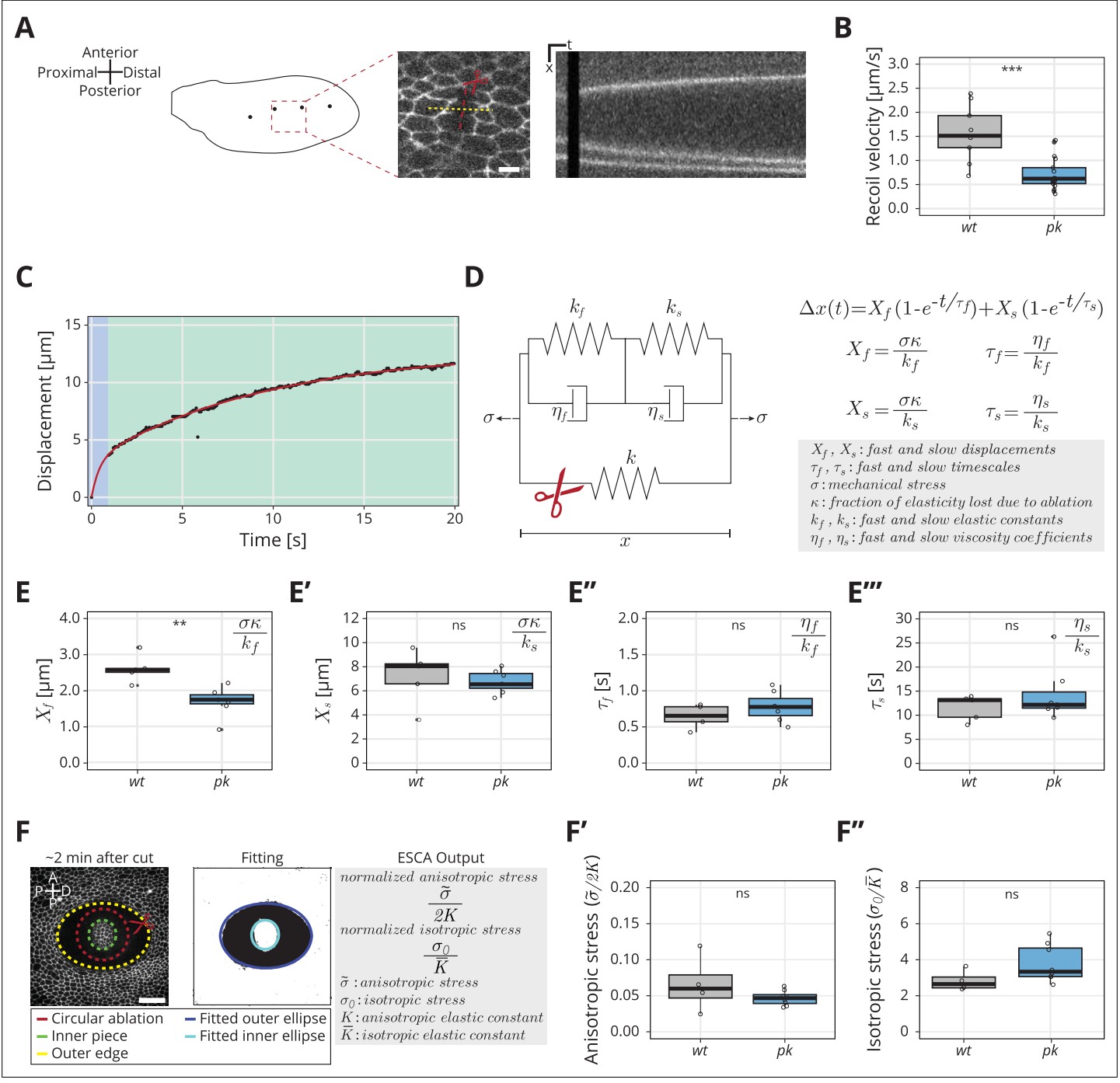

**Figure 2.** Rheological model for the response to laser ablation: (**A**) Schematic of a *wt* wing at −8 hRPCE. Linear laser ablation experiments were performed in the blade region enclosed by the red square. Dots on the wing cartoon indicate sensory organs. The red line corresponds to the ablation, and the kymograph was drawn perpendicularly to the cut (yellow). Scale bar, 5 μm. (**B**) Initial recoil velocity upon ablation (simplified as recoil velocity in the *y*-axis title) along the proximal–distal (PD) axis at −8 hRPCE for *wt* (gray) and *pk* (blue) tissues (*n* ≥ 9). Significance is estimated using the Mann–Whitney *U*-test. ***p-val ≤0.001. (**C**) Example of the measured displacement after laser ablation (black dots) and corresponding exponential fit of the mechanical model (red curve). The blue and green regions highlight the displacement in the fast and slow timescale, respectively. (**D**) Description of the mechanical model that was devised to analyze the tissue response upon laser ablation. After the cut, the spring with elastic constant *k* is ablated (red scissor), and the tissue response is given by the combination of the two Kelvin–Voigt models arranged in series. These two correspond to the fast response given by $k_f$ and $\eta_f$ and the slow response given by $k_s$ and $\eta_s$. The mechanical stress σ is constant. The membrane displacement $\Delta x(t)$ is calculated as a sum of the displacement ($X_f$) associated with the fast timescale ($\tau_f$) and the displacement ($X_s$) associated with the slow timescale ($\tau_s$). (**E–E'''**) Values obtained for each of the four fitting parameters when fit to the data. (**E**) Displacement associated with the fast and (**E'**) slow timescale

*Figure 2 continued*

for *wt* (gray) and *pk* (blue). (**E″**) Fast and (**E‴**) slow timescale for *wt* (gray) and *pk* (blue) ($n \geq 5$). Significance is estimated using the Student's *t*-test. **p-val ≤0.01; ns, p-val >0.05. (**F**) Example of a circular laser ablation used for analysis with elliptical shape after circular ablation (ESCA). The left image shows the final shape of the ablation around 2 min after cut, and the right image shows the corresponding segmented image, where the inner and outer pieces were fit with ellipses. After the fitting, the model outputs the anisotropic and isotropic stress (equations shown on the right side). Scale bar, 20 μm. A = anterior, P* = posterior, D = distal, P = proximal. (**F′**) Anisotropic stress $\widetilde{\sigma}/2K$ for *wt* (gray) and *pk* (blue) tissues at −8 hRPCE ($n \geq 4$). Significance is estimated using the Mann–Whitney *U*-test. ns, p-val >0.05. (**F″**) Isotropic stress $\sigma_0/\overline{K}$ for *wt* (gray) and *pk* (blue) tissues at −8 hRPCE ($n \geq$ 4). Significance is estimated using the Mann–Whitney *U*-test. ns, p-val >0.05. Time is relative to peak cell elongation (hRPCE). In all plots, each empty circle indicates one cut, and the box plots summarize the data: thick black line indicates the median; the boxes enclose the first and third quartiles; lines extend to the minimum and maximum without outliers, and filled circles mark outliers.

The online version of this article includes the following source data and figure supplement(s) for figure 2:

**Source data 1.** Numerical data for *Figure 2B*, initial recoil velocity upon ablation along the proximal–distal (PD) axis for *wt* and *pk* tissues.

**Source data 2.** Numerical data for *Figure 2E*, values for fitted parameters of the rheological model.

**Source data 3.** Numerical data for *Figure 2F–F″*, values for anisotropic and isotropic stress deteremined with elliptical shape after circular ablation (ESCA).

**Figure supplement 1.** Initial recoil velocity upon linear laser ablation for *stbm* and *fmi* mutant wings, exponential fits of cell response upon laser ablation, and ratio of elastic constants obtained by elliptical shape after circular ablation (ESCA) at −8 hRPCE.

**Figure supplement 1—source data 1.** Numerical data of *Figure 2—figure supplement 1A*.

**Figure supplement 1—source data 2.** Numerical data of *Figure 2—figure supplement 1B*.

**Figure supplement 1—source data 3.** Numerical data of *Figure 2—figure supplement 1C*.

circular ablation) (*Dye et al., 2021*), which uses circular laser ablation and quantifies the resulting elliptical tissue outline once the mechanical equilibrium is established (*Figure 2F* and Materials and methods, Elliptical shape after circular ablation). Analysis of the elliptical tissue outline provides information about two-dimensional stresses present in the tissue before the ablation. In particular, we measure the magnitude of the anisotropic shear stress tensor, normalized by the shear elastic modulus $\tilde{\sigma}/(2K)$ and the isotropic stress normalized by the area elastic modulus $\sigma_0/\overline{K}$. The stress $\sigma$ in the simple rheological model presented above would correspond to tissue stress normal to the linear laser ablation axis and therefore it is a linear combination of both $\tilde{\sigma}/(2K)$ and $\sigma_0/\overline{K}$. ESCA also provides an estimate of the ratio of shear and area elastic constants $2K/\overline{K}$.

Using ESCA, we find no significant difference between *wt* and *pk* mutants in anisotropic and isotropic stress magnitudes, nor in the ratio of elastic constants (*Figure 2F–F″* and *Figure 2—figure supplement 1C*). Since the ratio $\sigma/k_s$ defined in the rheological model is related to the normalized tissue stresses and elastic moduli, which do not change as shown by ESCA, we conclude that that $\sigma/k_s$ is not different between *wt* and *pk*. Therefore, we account for the observed changes of fast timescale displacement $X_f$ in the *pk* mutant with a change of the single elastic constant $k_f$. In this scenario, $\eta_f$ changes together with the $k_f$, such that $\tau_f = \eta_f/k_f$ is conserved. This suggests that fast elasticity and viscosity are not independent but stem from a microscopic mechanism that controls the relaxation timescale. An example of such mechanism is turnover of the acto-myosin network, although this mechanism would be too slow to account for the fast relaxation timescale we observe. The conclusion that only the short-time response to the ablation, and not the tissue stress, is affected in the *pk* mutant is consistent with the lack of a clear phenotype in the large-scale tissue flows (*Figure 1*).

## Dynamics of stress and cell elongation throughout blade elongation flows in wild type and *core PCP* mutants

To examine the effect of PCP mutation throughout blade elongation flows, we aimed to simplify the time intensive segmentation of the full ablation

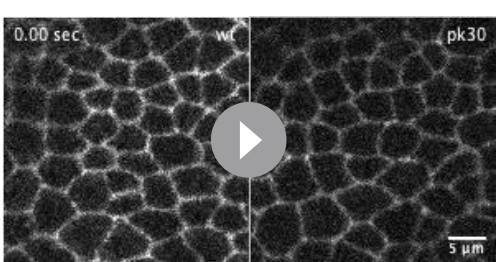

**Video 1.** Shown here is an example of a linear laser ablation, cutting three to four cells, in *wt* (left) or *pk* pupal wings. The movie goes dark during the ablation itself. Thereafter, the tissue displaces. Anterior is up; proximal is left.

https://elifesciences.org/articles/85581/figures#video1

dynamics. To this end, we measured only the initial recoil velocity at different developmental time-points. In terms of our model, the initial recoil velocity measured during first $\delta t = 0.65s$ can be expressed as $v = (X_f(1 - e^{-\delta t/\tau_f}) + X_s(1 - e^{-\delta t/\tau_s}))/\delta t$. Since the value of $\delta t$ is comparable to the fast timescale $\tau_f$, about 63% of the $X_f$ value relaxes over that time, while at the same time only about 5% of the $X_s$ value is relaxed. Using the measured values of $X_f$ and $X_s$, we estimate that the fast timescale dynamics contributes about 80% of the $v$ value. Therefore, the initial recoil velocity is a good proxy for the fast displacement $X_f$.

We find that the initial recoil velocity along the PD axis peaks at −8 hRPCE before declining again by 4 hRPCE (*Figure 3A*), consistent with previous work (*Iyer et al., 2019*). The behavior of the initial recoil velocity in the *pk* mutant is qualitatively similar throughout blade elongation flows, however, with significantly lower magnitude than *wt* (*Figure 3A*). We also observed this behavior in *stbm* and *fmi* mutant tissues (*Figure 3—figure supplement 1A*). This result indicates that $X_f$ is lower in *core PCP* mutants than in *wt* throughout blade elongation flows.

We also performed ESCA at different timepoints in *pk* mutants and observe that anisotropic stress ($\tilde{\sigma}/2K$) rises early during blade elongation flows before eventually declining (*Figure 3B*), whereas isotropic stress ($\sigma_0/\overline{K}$) remains fairly constant (*Figure 3B*). Strikingly, ESCA does not report any difference in measured stresses between *pk* and *wt*, nor in the ratio of elastic constants ($2K/\overline{K}$, *Figure 3—figure supplement 1B*) throughout blade elongation flows. To further compare the information contained in the initial recoil velocity with the anisotropic stress measured by ESCA, we performed linear ablations also in the perpendicular orientation. With such data, we could quantify the difference in initial recoil velocity between the two orientations $\delta v = v_{PD} - v_{AP}$, which is expected to be proportional to the shear stress along the PD wing axis. We then quantified how $\delta v$ evolves throughout the blade elongation flows (*Figure 3—figure supplement 1C, D*). Whereas ESCA clearly shows that stresses in the tissue remain the same in *wt* and *pk* throughout blade elongation flows, the difference in initial recoil velocity $\delta v$ is significantly lower in *pk* compared to *wt*. This result indicates that our conclusions based on the −8 hRPCE timepoint are true throughout blade elongation flows, namely that the differences in $X_f$ between *wt* and *pk* stem from the fast elastic constant $k_f$ and not from the differences in mechanical stresses in the tissue.

To further probe the possible role of core PCP in epithelial mechanics, we also measured the dynamics of the PD component of cell elongation ($Q$) in *wt* and *core PCP* mutants (*Figure 3C*, *Figure 3—figure supplement 2E*). Interestingly, in both *wt* and *pk*, anisotropic stress peaks around −6 hRPCE (*Figure 3B*), whereas $Q$ peaks significantly later, between −4 and 0 hRPCE. We have previously related the tissue stress and cell elongation through a constitutive relation $\tilde{\sigma} = 2KQ + \zeta$, where $\zeta$ represented an active anisotropic stress component (*Etournay et al., 2015*). The difference in timing of the peaks in stress and cell elongation indicate that the active stresses change over time. However, we observe no differences between *wt* and *core PCP* in the peak of cell elongation (*Figure 3D*, *Figure 3—figure supplement 2E*), showing that core PCP also does not affect active anisotropic stresses underlying the dynamics of cell elongation during blade elongation flows.

## Discussion

Here, we used the *Drosophila* pupal wing as a model for studying the interplay between planar polarized chemical signaling components, specifically the core PCP pathway, and the mechanical forces underlying tissue morphogenesis.

An extensive analysis of *core PCP* mutants shows no significant phenotype in pupal wing morphogenesis during the blade elongation flows. We find no significant differences in overall tissue shape change, nor in the pattern or dynamics of underlying cellular contributions. Even if a larger sample size of pupal wings would reveal a statistically significant phenotype, as indicated by our analysis of adult wings, the differences to the wild type would be subtle. Furthermore, we found no significant differences in tissue mechanical stress or in cell elongation over time. Generally these results are consistent in mutants that greatly reduce core PCP polarity (*stbm* and *fmi*) or prevent its decoupling from Fat (*pk*).

Interestingly, we do observe a phenotype in the initial recoil velocity upon laser ablation between *core PCP* mutants and *wt*, but this is not reflected in tissue stresses or large-scale morphogenetic flows that shape the wing. A detailed analysis of *wt* and *pk* suggests that the phenotype arises from a difference in the elastic constant $k_f$ underlying the fast timescale response ($\tau_f = 0.65s$) to the ablation. In

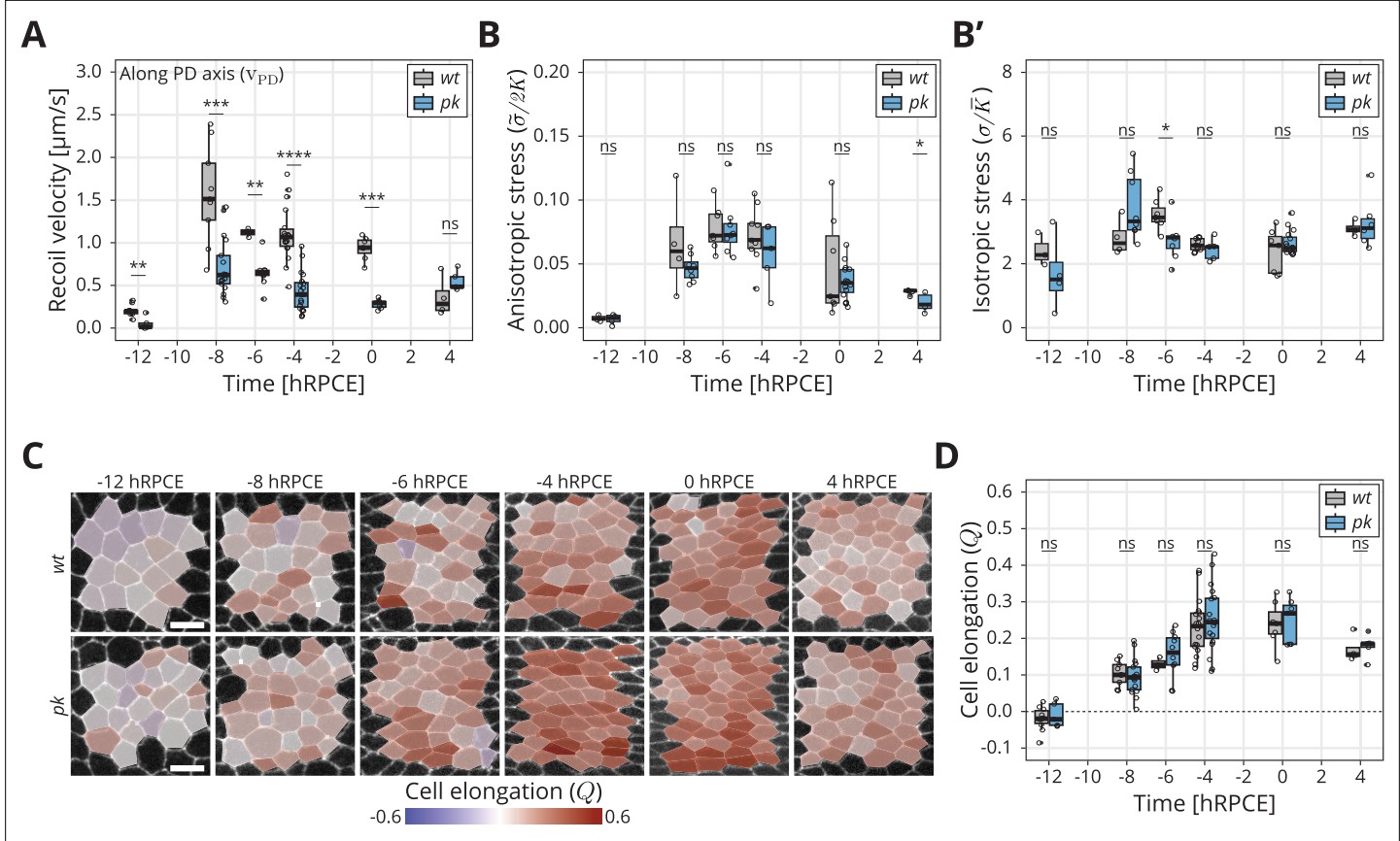

**Figure 3.** Dynamics of stress and cell elongation throughout blade elongation flows in *wt* and *pk* mutant. (**A**) Initial recoil velocity upon ablation (simplified as recoil velocity in the *y*-axis title) along the proximal–distal (PD) axis throughout blade elongation flows for *wt* (gray) and *pk* (blue) tissues (*n* ≥ 3). Significance is estimated using the Mann–Whitney *U*-test. \*\*\*\*p-val ≤0.0001; \*\*\*p-val ≤0.001; \*\*p-val ≤0.01; ns, p-val >0.05. (**B**) Elliptical shape after circular ablation (ESCA) results for anisotropic stress $\widetilde{\sigma}/2K$ for *wt* (gray) and *pk* (blue) tissues throughout blade elongation flows (*n* ≥ 3). Significance is estimated using the Mann–Whitney *U*-test. \*p-val <0.05; ns, p-val >0.05. (**B'**) ESCA results for isotropic stress $\sigma/\overline{K}$ for *wt* (gray) and *pk* (blue) throughout blade elongation flows (*n* ≥ 3). Significance is estimated using the Mann–Whitney *U*-test. \*p-val <0.05; ns, p-val >0.05. (**C**) Color-coded PD component of cell elongation $Q$ in the blade region between the second and third sensory organs found in the intervein region between L2 and L3. The images correspond to *wt* (top row) and *pk* (bottom row) wings throughout blade elongation flows. Scale bar, 5 μm. (**D**) Quantification of the PD component of cell elongation $Q$ in this region throughout blade elongation flows for *wt* (gray) and *pk* (blue) (*n* ≥ 3). Significance is estimated using the Mann–Whitney *U*-test. ns, p-val >0.05. Time is relative to peak cell elongation (hRPCE). In all plots, each empty circle indicates one experiment, and the box plots summarize the data: thick black line indicates the median; the boxes enclose the first and third quartiles; lines extend to the minimum and maximum without outliers, and filled circles mark outliers.

The online version of this article includes the following source data and figure supplement(s) for figure 3:

**Source data 1.** Numerical data for *Figure 3A*, initial recoil velocity upon ablation along the proximal–distal (PD) axis throughout blade elongation flows for *wt* and *pk* mutant tissues.

**Source data 2.** Numerical data for *Figure 3B–B'*, elliptical shape after circular ablation (ESCA) results for anisotropic and isotropic stress in *wt* and *pk* mutant tissues throughout blade elongation flows.

**Source data 3.** Numerical data for *Figure 3D*, proximal–distal (PD) component of cell elongation $Q$ throughout blade elongation flows for *wt* and *pk* mutant tissues.

**Figure supplement 1.** Study of pupal wing mechanics over time.

**Figure supplement 1—source data 1.** Numerical data of *Figure 3—figure supplement 1A*, initial recoil velocity upon ablation along the proximal–distal (PD) axis throughout blade elongation flows for wt, stbm, and fmi tissues.

**Figure supplement 1—source data 2.** Numerical data of *Figure 3—figure supplement 1B*, elliptical shape after circular ablation (ESCA) report of ratio of elastic constants throughout blade elongation flows for wt and pk.

**Figure supplement 1—source data 3.** Numerical data of *Figure 3—figure supplement 1C*, initial recoil velocity along the anterior–posterior (AP) axis for wt and pk.

**Figure supplement 1—source data 4.** Numerical data of *Figure 3—figure supplement 1D*, proxy for shear stress calculated as the difference

*Figure 3 continued*

between the initial recoil velocities along the proximal–distal (PD) and anterior–posterior (AP) axes for wt and pk, and elliptical shape after circular ablation (ESCA) report of anisotropic stress throughout blade elongation flows for wt and pk.

**Figure supplement 2.** Quantification of cell elongation in the blade region throughout blade elongation flows.

**Figure supplement 2—source data 1.** Numerical data of *Figure 3—figure supplement 2E*, proximal–distal (PD) component of cell elongation *Q* throughout blade elongation flows for wt, stbm, and fmi tissues.

our simple model, the fast and stiff spring $k_f$ has a small contribution to the effective tissue elasticity $\bar{k}$, which is dominated by the slow and soft spring $k_s$ (See Results, A rheological model for the response to laser ablation). The observation that core PCP only affects $k_f$ is therefore consistent with the lack of phenotype at larger scales. What is the the biophysical nature of the fast response to laser ablation? We hypothesize that processes that react on timescales <1 s to a laser ablation could be related to cortical mechanics of cell bonds or possibly changes in cell hydraulics, and it is unclear how core PCP would affect these processes. Whether this core PCP phenotype in $k_f$ leads to very subtle changes in tissue development not detected in our analyses here, or is only visible in response to a laser ablation, also remains unknown and would require a much larger sample size to address. For the adult wing, we have a sufficient sample size to reveal a weak but significant shape phenotype in *core PCP* mutant wings. This result suggests that a weak phenotype arises during pupal development that we could not reliably detect in our analysis of cell dynamics. We also cannot rule out the possibility that there is a compensating mechanism that prevents the phenotype from appearing at larger scales.

Initial recoil velocity after a laser ablation is often used a proxy for tissue mechanical stresses. However, our results highlight a limitation of this approach for looking at how stress changes in different genotypes, as here we show how initial recoil velocity is influenced by differences in mechanics on small scales that are not necessarily related to differences in overall tissue stress.

While we have shown that core PCP is not required to organize the dynamic patterns of cellular events underlying blade elongation flows, it might still affect later stages of wing development. Furthermore, there may still be other patterning systems acting redundantly or independently with core PCP. For example, the Fat PCP system and Toll-like receptors have been shown to influence the orientation of cellular rearrangements and cell divisions in other contexts (*Bosveld et al., 2012*; *Mao et al., 2006*; *Paré et al., 2014*; *Lavalou et al., 2021*, reviewed in *Umetsu, 2022*). Whether and how other polarity systems influence pupal wing morphogenesis remains unknown. Alternatively, anisotropic mechanical stress induced by hinge contraction could itself provide a polarity cue through mechano-sensitive activity of the cytoskeleton. Our recent work in the larval wing disc shows that the cell polarity that drives the patterning of cell shape and mechanical stress contains a mechano-sensitive component (*Dye et al., 2021*). Here, we show a detailed analysis of tissues stress dynamics and cell elongation in the pupal wing revealing that the active cellular stresses that are relevant for pupal wing morphogenesis (*Etournay et al., 2015*) change in time (*Figure 3*). Whether the same mechano-sensitive mechanism established in the larval wing can also account for the dynamics of active stresses during the pupal blade elongation flows will be an important question to answer in the future.

## Materials and methods

### Key resources table

| Reagent type (species) or resource | Designation | Source or reference | Identifiers | Additional information |
|---|---|---|---|---|
| gene (*Drosophila melanogaster*) | *w-* | NA | FLYB:FBal0018186 | |
| gene (*Drosophila melanogaster*) | *shg (shotgun; E-cadherin)* | NA | FLYB:FBgn0003391 | |
| gene (*Drosophila melanogaster*) | *pk30* | NA | FLYB:FBal0101223 | |
| gene (*Drosophila melanogaster*) | *stbm6* | NA | FLYB:FBal0062423 | |
| gene (*Drosophila melanogaster*) | *fmifrz3* | NA | FLYB:FBal0143193 | |

*Continued on next page*

*Continued*

| Reagent type (species) or resource | Designation | Source or reference | Identifiers | Additional information |
|---|---|---|---|---|
| strain, strain background (*Drosophila melanogaster*, male) | *wt* | Other | PMID:19429710 | *Huang et al., 2009*. Genotype: wt: w-; EcadGFP; |
| strain, strain background (*Drosophila melanogaster*, male) | *pk* | Bloomington *Drosophila* Stock Center | RRID:BDSC_44229 | *Gubb et al., 1999*. Genotype: w-; EcadGFP, pk30; |
| strain, strain background (*Drosophila melanogaster*, male) | *stbm* | Bloomington *Drosophila* Stock Center | RRID:BDSC_6918 | *Wolff and Rubin, 1998*. Genotype: w-; EcadGFP, stbm6; |
| strain, strain background (*Drosophila melanogaster*, male) | *fmi* | Bloomington *Drosophila* Stock Center | RRID:BDSC_6967 | *Wolff and Rubin, 1998*. Genotype: w-; EcadGFP, fmifrz3; |
| chemical compound, drug | Euparal | Carl Roth | 7356.1 | |
| chemical compound, drug | Holocarbon oil 700 | Sigma-Aldrich | H8898 | |
| chemical compound, drug | Isopropanol (2-propanol) | Sigma-Aldrich | 1.0104 | |
| software, algorithm | Fiji | Other | v. 2.0.0-rc-68/1.52e | *Schindelin et al., 2012* |
| software, algorithm | Ilastik | Other | v. 1.2.2 | *Berg et al., 2019* |
| software, algorithm | MATLAB | Other | v. 9.2.0.1226206 (R2017a) | *MATLAB, 2017* |
| software, algorithm | PreMosa | Other | | *Blasse et al., 2017* |
| software, algorithm | R | Other | v. 3.4.1 | *R Development Core Team, 2020* |
| software, algorithm | Rstudio | Other | v. 3.6.1 | *RStudio Team, 2020* |
| software, algorithm | TissueMiner | Other | v. TM_1.0.2 | *Etournay et al., 2016* |
| other | Coverslip | Paul Marienfeld GmbH | 107052 | |
| other | Microscope slides | Paul Marienfeld GmbH | 1000200 | |
| other | Dumont #55 Forceps | Fine Science Tools | 11295–51 | |
| other | Vannas Spring Scissors | Fine Science Tools | 15000–08 | |

## Fly husbandry

Flies were maintained at 25°C and fed with standard fly food containing cornmeal, yeast extract, soy flour, malt, agar, methyl 4-hydroxybenzoate, sugar beet syrup, and propionic acid. Flies were kept at 25°C in a 12-hr light/dark cycle. Vials were flipped every 2–3 days to maintain a continuous production of pupae and adult flies. All experiments were performed with male flies, since they are slightly smaller and therefore the wings require less tiling on the microscope to be imaged than females.

## Long-term timelapse imaging of pupal wing morphogenesis
### Acquisition
White male pupae were collected, slightly washed with a wet brush, and transferred to a vial containing standard food. At 16 hAPF, the pupal case was carefully dissected so that the wing would be accessible. The pupae was then mounted onto a 0.017-mm coverslip on a self-built metal dish with a drop of Holocarbon oil 700 (*Classen et al., 2008*). Pupal wing morphogenesis was imaged every 5 min for approximately 24 hr, as in *Etournay et al., 2015*. Wings that did not develop after 4–5 hr of imaging were discarded and not analyzed.

Two different microscopes were used for acquisition of long-term timelapses. All *wt*, *pk*, and *stbm* movies were acquired using a Zeiss spinning disk microscope driven by ZEN 2.6 (blue edition). This microscope consists of a motorized XYZ stage, an inverted stand, a Yokogawa CSU-X1 scan head, and a temperature-controlled chamber set to 25°C. The sample was illuminated with a 488-nm laser, and

**Table 1.** Date of acquisition of all long-term timelapses.

| Genotype | Date of acquisition | Start [hAPF] | End [hAPF] |
|---|---|---|---|
| | March 30, 2016 | 16 | 39.83 |
| | April 2, 2016 | 16 | 36.58 |
| | April 3, 2016 | 16 | 36.50 |
| | April 13, 2016 | 16 | 32.58 |
| wt | June 20, 2018 | 16 | 41.17 |
| | April 9, 2016 | 16 | 39.00 |
| | June 28, 2016 | 16 | 36.00 |
| pk | June 29, 2016 | 16 | 39.92 |
| | November 25, 2015 | 16 | 40.67 |
| | November 28, 2015 | 16 | 35.33 |
| stbm | December 11, 2014 | 16 | 37.58 |
| | October 20, 2018 | 16 | 39.17 |
| fmi | July 20, 2019 | 16 | 38.17 |

the emission was collected using a 470/40 bandpass filter, through a Zeiss 63 × 1.3 W/Gly LCI Plan-Neofluar objective and a Zeiss AxioCam Monochrome CCD camera with 2 × 2 binning. The whole wing was imaged in 24 tiles with an 8 % overlap. Each tile consisted of 50–60 stacks with a Z-spacing of 1 μm. The laser power was set to 0.1 mW.

The two *fmi* movies were acquired with an Olympus IX 83 inverted stand driven by the Andor iQ 3.6 software. The microscope is equipped with a motorized xyz stage, a Yokogawa CSU-W1 scan head, and an Olympus 60 × 1.3 Sil Plan SApo objective. The setup was located inside a temperature-controlled chamber set to 25°C. The sample was illuminated with a 488-nm laser, and the emission was collected using a 525/50 bandpass filter. The whole wing was imaged by tiling with eight tiles with a 10 % overlap. Each tile consisted of 50–60 stacks with a distance of 1 μm between them. The laser power was set to 0.75 mW.

*Table 1* summarizes the date when the long-term timelapses were acquired and the age of the pupae during the imaging.

## Processing, segmentation, tracking, and database generation

Raw stacks were projected, corrected for illumination artifacts, and stitched using PreMosa (*Blasse et al., 2017*). The stitched images of individual timepoints were cropped to fit the wing size, registered using the Fiji plugin 'Descriptor-based series registration (2D/3D + t)', and converted to 8 bit with Fiji (*Schindelin et al., 2012*). The segmentation was performed with the Fiji plugin TissueAnalyzer (*Schindelin et al., 2012*; *Aigouy et al., 2010*; *Aigouy et al., 2016*). Segmentation errors were identified and manually corrected by looking at the cell divisions and deaths masks.

Subsequent processing and quantifications were performed using TissueMiner (*Etournay et al., 2016*). Before generating the relational database, we rotated the movies so that the angle formed by a manually drawn line connecting the sensory organs would be 0. We manually defined the regions of interest, such as the blade, hinge, and the anterior and posterior regions, using the last frame of the movie. Next, we generated the relational database containing information about the cellular dynamics during blade elongation flows using TissueMiner (*Etournay et al., 2016*).

We queried and worked with the data using the Dockerized version of RStudio (*Nickoloff, 2016*), which loads all packages and functions required to work with TissueMiner. Movies were aligned by the peak of cell elongation by fitting a quadratic function around the cell elongation values 40 frames before and after the absolute maximum of cell elongation in the blade region for each movie. The maximum of this curve was identified and set as the timepoint 0 hRPCE.

## Adult wing preparation and analysis of wing shape

Adult male flies were fixed in isopropanol for at least 12 hr. One wing per fly was dissected in isopropanol, transferred to a microscope slide and covered with 50% euparal in isopropanol. Wings were mounted with 50–70 µl 75 % euparal/isopropanol.

*wt*, *pk*, and *stbm* wings were imaged using a Zeiss widefield Axioscan Z1 microscope equipped with a Zeiss 10 × 0.45 air objective. *fmi* wings were imaged using a Zeiss widefield Axiovert 200 M microscope equipped with a Zeiss 5 × 0.15 Plan-Neofluar air objective.

Wing blade parameters were quantified using a custom-written Fiji macro (provided as *Source code 1*) (*Schindelin et al., 2012*). The shape or major-to-minor ratio was calculated using a custom RStudio script (*R Development Core Team, 2020*; *RStudio Team, 2020*).

## Subsampling and statistical analysis

Random sampling was done using a custom written RStudio pipeline (*R Development Core Team, 2020*; *RStudio Team, 2020*). A group of a given sample size was randomly selected with replacement for each group (*wt*, *pk*, *stbm*, and *fmi*), and a Kruskal–Wallis test was ran to compare them. This analysis was repeated 10,000 times. The sample sizes analyzed were 3, 4, 5, 6, 7, 8, 9, 10, 20, and 40.

## Quantification of the PD component of cell elongation $Q$

Prior to all laser ablation experiments, we acquired a stack of 50 µm thick that was projected using PreMosa (*Blasse et al., 2017*). We cropped a region that enclosed the region that was ablated, segmented cells using TissueAnalyzer (*Aigouy et al., 2010*; *Aigouy et al., 2016*), and generated a relational database with TissueMiner (*Etournay et al., 2016*).

The definition of cell elongation was first presented in *Aigouy et al., 2010* and it describes the angle and magnitude of the tensor. The cell elongation tensor is given by

$$\begin{pmatrix} \epsilon_{xx} & \epsilon_{xy} \\ \epsilon_{xy} & -\epsilon_{xx} \end{pmatrix}, \tag{2}$$

where

$$\epsilon_{xx} = \frac{1}{A_c} \int cos(2\phi) \, dA \tag{3}$$

and

$$\epsilon_{xy} = \frac{1}{A_c} \int sin(2\phi) \, dA. \tag{4}$$

Cell elongation is normalized by the cell area ($A_c$) of each cell. The magnitude of cell elongation is:

$$\epsilon = (\epsilon_{xx}^2 + \epsilon_{xy}^2)^{\frac{1}{2}} \tag{5}$$

Here, we plot $\epsilon_{xx}$ as $Q$, which we describe as the PD component of cell elongation.

**Table 2.** Parameters used to perform laser ablations.

|  | Linear ablations | Circular ablations |
| --- | --- | --- |
| Exposure time [s] | 0.05 | 0.05 |
| 488-nm laser intensity [%] | 50 | 50 |
| Time interval [s] | 0.09 | 2.55 |
| Pulses per shot | 25 | 25 |
| Shots per µm | 2 | 2 |
| Shooting time [s] | 0.67 | 147.28 |
| Thickness of stack ablated [µm] | 1 | 20 |

## Laser ablation experiments

Pupae were dissected and mounted as described for the long-term timelapses. Ablations were always performed in the same region of the wing blade, found in the intervein region between the longitudinal veins L3 and L4 and between the second and third sensory organs. This region was chosen because these landmarks are easily visible in all timepoints. Laser ablations were performed using a Zeiss spinning disk microscope equipped with a CSU-X1 Yokogawa scan head, an EMCCD Andor camera, a Zeiss 63 × 1.2 water immersion Korr UV-VIS-IR objective, and a custom-built laser ablation system using a 355-nm, 1000-Hz pulsed ultraviolet (UV) laser (*Grill et al., 2001*; *Mayer et al., 2010*). The imaging and cutting parameters for line and circular laser ablations are shown in *Table 2*. All laser ablation experiments were performed between January 2018 and July 2020, after the delay in pupal wing morphogenesis was identified.

### Linear laser ablations to calculate the initial recoil velocity

We performed both types of linear ablations in only one plane of the tissue, in order to minimize the time required for ablation and therefore be able to acquire the initial recoil velocity upon ablation (no imaging is possible during ablation). The length of the linear laser ablations was 10 μm, ablating three to four cells. We drew kymographs perpendicularly to the cut to follow the two edges of one ablated cell using Fiji (*Schindelin et al., 2012*). The initial recoil velocity was calculated as the average displacement of two membranes of the same cell that occurred during the black frames of the ablation itself. This calculation was made using a self-written MATLAB script (*MATLAB, 2017*). Scripts used to make kymographs and analyze the laser ablations are provided in *Source code 2*; *Source code 3*; *Source code 4*. The image acquired prior to the laser ablation was used to compute $Q$ in that region, as described in Quantification of the PD component of cell elongation $Q$, and the time corresponding to the maximum of cell elongation was defined as 0 hRPCE.

### Elliptical shape after circular ablation

Circular laser ablations used for ESCA were 20 μm in radius (approximately 10 cells). This radius was selected such that it would fit into the same blade region throughout the blade elongation flows. Due to the bigger size of these cuts and the curvature of the tissue, we cut the tissue along a stack of 20 μm thick. Approximately 2 min after the ablation, we acquired a stack of 50 μm. This image was projected using PreMosa (*Blasse et al., 2017*) and preprocessed by applying Gausian blur ($\sigma = 1$) and background subtraction filters (rolling ball radius = 30) in Fiji (*Schindelin et al., 2012*). The next steps were performed as in *Dye et al., 2021*: the image of the final shape of the cut was segmented using Ilastik (*Berg et al., 2019*) by defining three regions: membrane, cell, and dark regions. The segmented image was thresholded to obtain a binary image of the final shape of the cut. We fitted two ellipses to this image: one to the inner piece and another one to the outer outline of the cut. Based on the shape of these ellipses, the method outputs the anisotropic $\frac{\tilde{\sigma}}{2K}$ and isotropic stress $\frac{\sigma}{K}$ as a function of their respective elastic constants, and the ratio of elastic constants $\frac{2K}{\tilde{K}}$. A small number of experiments were fitted poorly (defined as an error per point greater than 0.3) and were therefore excluded from analysis. Prior to the circular ablation, a stack of 50 μm was acquired and used to calculate cell elongation before ablation (Quantification of the PD component of cell elongation $Q$). The time corresponding to the maximum of cell elongation was set to be 0 hRPCE.

## Kymograph analysis and fit to model

The ablations used to calculate the mechanical stress along the PD axis for *wt* and *pk* were further analyzed with the rheological model. To do so, we processed the kymographs by applying a Gaussian blur ($\sigma = 1$) (*Schindelin et al., 2012*), and then we segmented these kymographs with Ilastik (*Berg et al., 2019*). Using a self-written Fiji macro (*Schindelin et al., 2012*), we extracted the intensity profile for each timepoint. Next, we wrote an R script (*R Development Core Team, 2020*; *RStudio Team, 2020*) to identify the membrane displacement over time and obtained a unique curve per kymograph, which could be fitted with our model. We modeled a local patch of tissue as a combination of a spring with spring constant $k$, representing the ablated cells, and two KV elements with spring constants $k_f$ and $k_s$ and viscosity coefficients $\eta_f$ and $\eta_s$, representing the unablated cells, as shown in *Figure 2C, D*. Because the local tissue strain in the experimental measurement is expressed by the displacement of

the bond nearest to the ablation, in the rheological model we represent tissue strain by displacements of the two KV elements. In principle, the strain can be recovered by normalizing the displacements by the width of ablated cells. Displacements of the two KV elements are defined as a change in the distance between the end points of the KV elements $x_i(t)$, relative to their initial values $x_i(0)$, where $i \in \{f, s\}$ for fast ($f$) and slow ($s$) element.

Mechanical stress in the tissue is represented by the $\sigma$ acting on our model, and we assume that $\sigma$ is not changed by the ablation. Before the ablation, the model is in mechanical equilibrium and we can write

$$\sigma = (k + \bar{k})x(0), \tag{6}$$

where $x(0)$ is the initial distance between the two end points of the model, and $\bar{k} = k_f k_s / (k_f + k_s)$ is the elastic constant of the two KV elements connected in series. Upon ablation, the spring $k$ is removed and stresses in the model are imbalanced. The distance between the end points of the model $x(t)$ then evolves toward the new equilibrium position. The distance $x(t)$ can be decomposed as $x(t) = x_f(t) + x_s(t)$, where $x_f(t)$ and $x_s(t)$ are the time-dependent distances between end points of the two KV elements, representing their strains. The dynamics of $x(t)$ is then obtained by writing the force balance equation for the two KV elements

$$\sigma = k_f x_f(t) + \eta_f \frac{dx_f(t)}{dt}, \tag{7}$$

$$\sigma = k_s x_s(t) + \eta_s \frac{dx_s(t)}{dt}, \tag{8}$$

We solve for $x_f(t)$ and $x_s(t)$ to obtain

$$x_f(t) = \frac{\sigma}{k_f}(1 - e^{-t/\tau_f}) + x_f(0)e^{-t/\tau_f}, \tag{9}$$

$$x_s(t) = \frac{\sigma}{k_s}(1 - e^{-t/\tau_s}) + x_s(0)e^{-t/\tau_s}, \tag{10}$$

where

$$x_{f,s}(0) = \frac{\sigma(1 - \kappa)}{k_{f,s}}, \tag{11}$$

and $\kappa = k/(k + \bar{k})$ is the fraction of the overall model elasticity $k + \bar{k}$ destroyed by the ablation. The displacement relative to the initial configuration $\Delta x(t) = x(t) - x(0)$ is therefore

$$\Delta x(t) = X_f \left(1 - e^{-t/\tau_f}\right) + X_s \left(1 - e^{-t/\tau_s}\right), \tag{12}$$

where we introduced the long time displacements associated with the two KV elements

$$X_{f,s} = \frac{\sigma \kappa}{k_{f,s}}. \tag{13}$$

For simplicity, in the main text we refer to the long time displacements $X_f$ and $X_s$ of the two KV elements simply as displacements.

## Statistical analysis

Statistical analysis was done using R (*R Development Core Team, 2020*; *RStudio Team, 2020*). We first tested normality of the data using the Shapiro–Wilk test. When data were normal, we used Student's *t*-test to test statistical significance between two groups and analysis of variance test for multiple groups. When data were not normally distributed, significance was tested using the Mann–Whitney *U*-test for two groups and Kruskal–Wallis test for multiple groups. Statistical test results are shown on the figure captions.

## Acknowledgements

We thank Stephan Grill for giving us access to the microscope used for laser ablation. We thank the Light Microscopy Facility, the Computer Department, and the Fly Keepers of the MPI-CBG for their support and expertise. We would like to thank Christian Dahmann and Jana Fuhrmann for comments on the manuscript prior to publication. This work was funded by Germany's Excellence Strategy – EXC-2068-390729961– Cluster of Excellence Physics of Life of TU Dresden, as well as grants awarded to SE from the Deutsche Forschungsgemeinschaft (SPP1782, EA4/10-1, EA4/10-2) and core funding of the Max-Planck Society to SE and NAD. NAD additionally acknowledges funding from the Deutsche Krebshilfe (MSNZ P2 Dresden). AK and RPG were funded through the Elbe PhD program. FSG was supported by a DOC Fellowship of the Austrian Academy of Sciences. CD acknowledges the support of a postdoctoral fellowship from the LabEx 'Who Am I?' (ANR-11-LABX-0071) and the Université Paris Cité IdEx (ANR-18-IDEX-0001) funded by the French Government through its 'Investments for the Future'. We dedicate this work to our coauthor Prof. Dr. Suzanne Eaton, who tragically passed away before the finalization of the project.

## Additional information

### Funding

| Funder | Grant reference number | Author |
|---|---|---|
| Max Planck Society | | Franz S Gruber<br>Abhijeet Krishna<br>Charlie Duclut<br>Carl D Modes<br>Frank Jülicher<br>Natalie A Dye<br>Suzanne Eaton<br>Romina Piscitello-Gómez<br>Marko Popović |
| Deutsche Forschungsgemeinschaft | EXC-2068-390729961 | Natalie A Dye |
| Deutsche Forschungsgemeinschaft | SPP1782/EA4/10-1 | Suzanne Eaton<br>Natalie A Dye<br>Romina Piscitello-Gómez<br>Franz S Gruber |
| Deutsche Krebshilfe | MSNZ-P2 Dresden | Natalie A Dye |
| Austrian Academy of Sciences | DOC Fellowship | Franz S Gruber |
| Agence Nationale de la Recherche | ANR-11-LABX-0071 | Charlie Duclut |
| Agence Nationale de la Recherche | ANR-18-IDEX-0001 | Charlie Duclut |
| Deutsche Forschungsgemeinschaft | SPP1782/EA4/10-2 | Suzanne Eaton<br>Natalie A Dye<br>Romina Piscitello-Gómez |

 The funders had no role in study design, data collection, and interpretation, or the decision to submit the work for publication.

### Author contributions

Romina Piscitello-Gómez, Data curation, Formal analysis, Investigation, Software, Validation, Visualization, Writing – original draft, Writing – review and editing; Franz S Gruber, Data curation, Formal analysis, Investigation, Software, Validation, Writing – review and editing; Abhijeet Krishna, Formal analysis, Methodology, Software, Writing – review and editing; Charlie Duclut, Formal analysis, Software; Carl D Modes, Validation, Investigation, Formal analysis, Software; Marko Popović, Investigation, Formal analysis, Data curation, Software; Frank Jülicher, Visualization, Validation, Investigation, Writing – original draft, Formal analysis, Writing – review and editing, Software; Natalie A Dye, Data

curation, Supervision, Writing – original draft, Writing – review and editing; Suzanne Eaton, Visualization, Validation, Investigation, Writing – original draft, Formal analysis, Writing – review and editing

### Author ORCIDs
Franz S Gruber ![orcid] https://orcid.org/0000-0003-2008-8460
Abhijeet Krishna ![orcid] https://orcid.org/0000-0002-9291-500X
Charlie Duclut ![orcid] https://orcid.org/0000-0002-8595-6815
Frank Jülicher ![orcid] http://orcid.org/0000-0003-4731-9185
Natalie A Dye ![orcid] https://orcid.org/0000-0002-4859-6670

### Decision letter and Author response
Decision letter https://doi.org/10.7554/eLife.85581.sa1
Author response https://doi.org/10.7554/eLife.85581.sa2

---

## Additional files

### Supplementary files
• MDAR checklist

• Source code 1. Fiji macro used to quantify size and shape of adult wings. Inputs raw image of an adult wing and outputs text document containing quantifications of area, perimeter, major axis length, minor axis length, and other measurements not used in this manuscript.

• Source code 2. Fiji macro used to draw kymographs from a laser ablation experiment. Inputs stack of images from a timelapse laser ablation experiment. Outputs kymograph image that is later used to compute the initial recoil velocity upon ablation (*Source code 3*).

• Source code 3. Matlab script used to calculate the initial recoil velocity upon laser ablation in linear cuts. Inputs include the path to a folder containing the kymograph for each cut, as well as the pixel size in microns and time interval between image acquisition. Outputs a mat file containing the initial recoil velocity calculated as the average between the recoil velocities of the two membranes of the ablated cell.

• Source code 4. Matlab script used to concatenate all calculated initial recoil velocities for a given dataset. Inputs the path to a folder containing the mat files output from first script (*Source code 3*). Outputs a list of recoil velocities for each analyzed laser ablation experiment.

### Data availability
Source data and code are provided for each figure.

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

## Appendix 1

During the course of this work, we identified a delay in the onset of blade elongation flows compared to previous work (*Appendix 1—figure 1A*; *Etournay et al., 2015*; *Piscitello-Gómez et al., 2023*). In the past, cells reached their maximum of cell elongation at 22.9 ± 0.4 hAPF, while now they reach it at 28 hAPF. Although we do not know the cause of this delay, we have ruled out differences in temperature (either on the microscope or during development), nutrition (plant vs yeast-based foods), genetic background, presence of the parasite *Wolbachia*, or circadian gating (*Piscitello-Gómez et al., 2023*). To deal with this variation and combine data acquired over the years, we present cell dynamics data aligned in time by the peaks of cell elongation, and we refer to this timepoint as 0 hRPCE (relative to peak cell elongation) (*Appendix 1—figure 1A*). We investigated the cell dynamics underlying blade elongation flows in the delayed flies and observed that the shear rates were comparable with the older flies (*Appendix 1—figure 1B*). Thus, it is reasonable to shift the curves by aligning them to a new reference time.

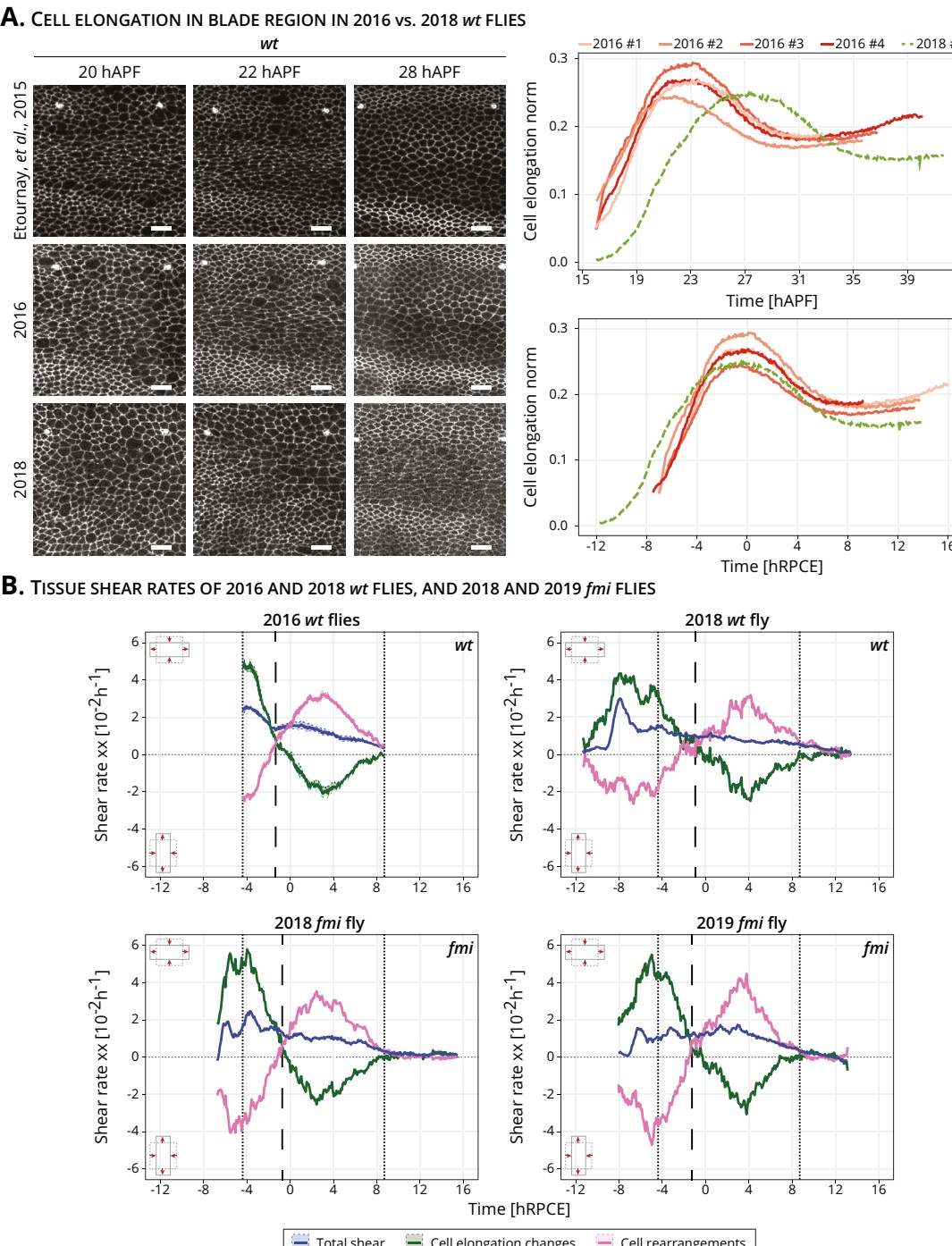

**Appendix 1—figure 1.** Delay and time alignment of old and newer flies: (**A**) Left: Snapshots of the blade region of long-term timelapses of *wt* pupal wing morphogenesis acquired in different years. Scale bar, 10 μm. Right: Cell elongation norm during blade elongation flows for old flies (orange palette, 2016 flies) and new flies (green curve, 2018 fly). Top plot: Cell elongation magnitude for each movie not aligned in time. The peak of cell elongation is delayed from around 23 to 28 hAPF. Bottom plot: Cell elongation magnitude after alignment in time to the peak of cell elongation. Time is now expressed in hours relative to peak cell elongation (hRPCE). (**B**) Cell dynamics underlying anisotropic tissue deformation for 2016 *wt* (*n* = 4, top left), 2018 *wt* flies (*n* = 1, top right), and 2 *fmi* flies imaged in 2018 (bottom left) and 2019 (bottom right). The vertical dashed line marks the timepoint where cell rearrangements flip from AP- to PD-oriented per movie. The two dotted lines mark the start and the end of the analyzed *wt* long-term timelapses acquired in 2016. The time is relative to the peak of cell elongation (hRPCE).

