## [Editor Report]

This valuable study shows that the core PCP pathway, which commonly orients morphogenetic processes in development, does not establish global cues for cellular movements in the *Drosophila* pupal wing. Using a combination of laser-ablation experiments and mathematical modelling, it provides compelling evidence that the core PCP pathway only affects the fast timescale cellular response, but does not appear to drive overall tissue dynamics. While the signalling pathway guiding this morphogenetic process remains to be elucidated, these relevant findings challenge the role of the core PCP pathway in morphogenesis.

---

## [Decision Letter]

**Decision letter after peer review:**

Thank you for submitting your article "Core PCP mutations affect short time mechanical properties but not tissue morphogenesis in the *Drosophila* pupal wing" for consideration by *eLife*. Your article has been reviewed by 3 peer reviewers, including Marcos Nahmad as Reviewing Editor and Reviewer #1, and the evaluation has been overseen by Michael Eisen as the Senior Editor. The following individual involved in the review of your submission has agreed to reveal their identity: M. Lisa Manning (Reviewer #3).

Essential revisions:

1) Please revise the text to be more clear about how you reconcile your observations that PCP mutants have an effect on the local tissue dynamics (e.g., recoil velocity upon laser ablation) but this has no global consequences on wing morphogenesis. In particular, could the local effects explain the subtle, but significant effects on adult wing shapes in these mutants?

2) Please justify your choice of pk30 mutants for most of the analysis, instead of using stbm6 or fmi mutants that completely disturb the core-PCP pathway.

3) Please explain in your rheological model why you believe that the recoil velocity has two phases.

4) Please reconsider the way you plot your data so that readers do not need to extract subtle differences on curves plotted in different graphs with different scales.

4) Please revise the discussion on the ESCA analysis, as reviewers have a hard time following it.

*Reviewer #1 (Recommendations for the authors):*

Overall the manuscript is well written and the results support the authors' conclusions; therefore I am supportive of publication. Here, I list a few comments that will improve the manuscript's readability, but will likely not affect the conclusions of the study:

– Considering that pk30 mutants maintain the polarity towards the margin i.e., both core-PCP and Fat-PCP remain aligned (Figure S1.1), while stbm6 and fmi mutants completely disturb the core-PCP pathway, why did the authors focus their analysis in the pk30 mutants (line 118)? If the purpose of the study is to examine the perturbations of the core-PCP, why not choose stbm6 or fmi mutants? The authors should justify this further.

– While it is clear that this study only focuses on the core-PCP pathway, perturbing only this pathway will not eliminate polarity and therefore it is not so surprising that oriented cell rearrangements persist. The authors should discuss which experiments (mutants) they suggest to perturb polarity altogether.

– The authors chose a specific region of the wing (between veins L3 and L4) to perform their perturbations. The justification is that this is a region that is easy to identify. But, do the results depend on this choice? Isn't the strength of the polarity signal larger closer to the wing margin?

– Use consistent English spelling throughout the manuscript (e.g., behaviour or behavior lines 169 vs. 171).

*Reviewer #2 (Recommendations for the authors):*

The authors have done a good job of trying to bring together what appear to be datasets generated over a fairly long period by different authors.

I do have a few issues regarding the presentation and interpretation of the data (specific points noted below). The main concern is that the authors keep flipping between saying core PCP has no effect on wing morphogenesis, to saying it has subtle effects to saying it has effects but they are not statistically significant. This is hard to reconcile with the convincing effect on tissue mechanics reveals by the laser ablation work, and possibly also the changes in adult wing shape and known roles of core PCP in both the wing and other tissues in the fly. Mostly I think it's just a question of careful choice of words and most likely leaving the question of effects of core PCP on morphogenesis open until all the data is considered together in the Discussion e.g. try to stick to observations in the Results and try to only do interpretation in the Discussion.

I also have queries about the rheological model and why the authors believe the recoil has 2 phases based on the data they show, and also about the way developmental timings have been recalibrated.

Specific points:

The references on lines 21 and 22 are an eclectic selection. The sentence seems to be about how polarized PCP proteins orient structures in epithelia, but the references include a mix of reviews and papers, some from before the polarization had been reported. For instance, Eaton 1997 was a seminal review that suggested planar polarization of stereocilia might be under the control of planar polarity pathways, but does not report such a result (I think the first papers were Montcouquiol et al. 2003 and Curtin et al. 2003?). It would probably be better for readers if the authors either cited reviews or if they want to cite primary papers, than just the first reports of polarized localization determining the orientation of structures (e.g. I think Usui et al. 1999 is the first in flies? Then Axelrod and Strutt?).

The references on lines 27 and 28 are mostly referring to Fat-Dachsous while the rest of the paragraph is about "core". There are papers about core PCP in flies affecting cell rearrangements/divisions. Segalen et al. 2010 is a good example, but not the first to report core PCP controlling SOP division orientation (Gho and Schweisguth 1998?). Sugimura and Ishihara 2013 report changes in cell rearrangements in the wing of a core PCP mutant. Others (moving away from the wing/notum) might include many papers about ommatidial rotation in the eye, joint formation in the legs, cell rearrangements in the tracheal system (and probably others). A better review of core PCP roles in tissue morphogenesis in flies would put the work in a better context.

In general, I would recommend thinking carefully about the references in the Introduction so that they are of the most help to readers from outside the field. It is hard to see an organizing principle behind the mixture of reviews from different decades, primary papers (but often not first reports), cloning, or genetic or biochemical characterization studies. As above, I would recommend recent reviews (where available) and primary papers that are the first reports, for each point the authors wish to make.

Line 44 – Fz is not a cadherin.

Figure S1A -I don't fully agree that 30 hAPF is the "End of morphogenesis" as suggested by the figure label. Even if you ignore the emergence of hairs and wing ridges, the epithelium itself will go on expanding and folding in subsequent hours. Similar statements are made on line 73 and on Figure 1.

Figure S1A legend – do "tissue flows reorganize Fat PCP"? My guess is Fat PCP continues to follow the Ds and Fj gradients and its insensitivity to the tissue flows is why it ends up perpendicular to core PCP. Of course, tissue flow might also have an effect but do the authors have evidence for this (I'm not sure Merkel et al. show this?). The statement in the legend is also not the same as what lines 51-52 say, where the message is that Fat PCP remains margin oriented.

Lines 66 and 67 – in normal nomenclature pk30 = pk[30] (i.e. 30 in superscript) and stbm6 = stbm[6], consistent with fmi[frz3]. I think it would be simplest to refer to stbm[6] as stbm and pk[30] as pk after the first definition, in the same way, fmi[frz3] is referred to as fmi, but this is of course up to the authors. As an aside, fmi[frz3] was described as a strong hypomorph in Chae et al. 1999, so it seems unlikely that in frz3/frz3 the "PCP network is absent", as stronger phenotypes are seen for frz3/Df, implying some residual activity.

Figure S1.2C – typo in the first word of the label.

Line 84 "these subtle changes are not statistically significant". This all looks reasonable, but possibly the authors need to qualify this by saying something like "considering the small numbers of wings analyzed". I appreciate the work involved to get these datasets was enormous, so higher n numbers are not feasible. However, with larger n numbers there might be a difference. The average shear values for all three mutant conditions are lower than wt. Ideally, a power calculation would be done to place a numerical value on the likelihood of the shear being the same in each condition, but again I appreciate this is difficult and possibly not appropriate post hoc.

Line 85 "also do not show differences" – see the previous point. Without stats, you can't say they don't differ, but here it is implied that the data in Figure S1.2C and D have been tested for differences and none was found. This is true for S1.2C with the caveat of small n numbers and no power calculation, but for S1.2D appears to be a judgment made by eye. However, by eye total tissue area at -2h looks lower for pk[3] and stbm[6] than for wt, and at e.g. 6h all 3 mutants show lower cell area changes than wild-type. I think the most the authors can say (unless I've missed something) is that the patterns of tissue behavior look broadly the same for the mutants as for wt, suggesting no gross differences. Note lines 89-90 say "Again, we find only subtle differences" which is not the same as no differences.

Lines 94-95 and Figure S1.4G – again I think it is deceptive to put weight on statistics with low n numbers and no power calculation. I'm not confident that you can "conclude that core PCP does not guide the global patterns of cell dynamics during pupal morphogenesis". The best that you can conclude is that changes are at best subtle (and in the experiments presented might be due to "noise" but could also be due to roles for core PCP in cell behavior). The authors rather undermine themselves by immediately going on to say that there is a reproducible difference in adult wing shape, although admittedly we don't know that this is due to events between 16-32h as the authors note. I think more nuanced conclusions would be better.

Lines 115-116 "cellular dynamics is [are?] basically unperturbed" is another example of wording that implies the authors think cellular dynamics are perturbed to some extent, but they want to minimize the significance of this observation. Similarly, lines 198-199 "no significant phenotype" belies the difficulty in making such an assertion when the genotypes do look different in some aspects and the tests have (apparently) minimal statistical power.

Lines 119-120 "When plotting displacement of the nearest bond to ablation over time, we observe both a fast (<1 s) and a slow (<20 s) regime (Figure 2C)." I think the authors are saying that their data fits a 2 phase exponential with a fast and slow phase, but I'm a bit nervous about this as the first phase seems to be based on a single data point at 1 sec. I'm not an expert on curve fitting, but only having 1 point to demonstrate the presence of a "fast" phase seems a bit unsafe. I couldn't find anything in the manuscript that describes why the authors decided a 2 phase fit was the correct choice. The subsequent model appears to assume 2 phases, but I'm interested to know if you can get just as good a fit with 1 phase and if you build a model with 1 phase does that still show pk30 having a significantly lower displacement over time? I'm concerned that a 1 phase model may have been unnecessarily ruled out.

Appendix 1 – I understand the issue confronting the authors. Earlier studies from the lab (from Aigouy et al. 2010 onwards) consistently saw peak cell elongation at ~23h, then in Iyer et al. peak elongation appears to be at ~28h (I'm unsure if Iyer et al. comments on this). I have a couple of comments:

Line 519 the statement "we identified a delay on the onset of pupal wing morphogenesis" is unclear to me. Do the authors mean that morphogenesis proceeds at a normal rate but just start late (similar to a train being delayed but running at the normal speed), or is everything just happening slower? The curves in panels A and B really look like the latter (e.g. in panel B the same sequence of events in 2016 takes 12h while in 2018 it takes 20h), but the authors seem to be saying it's the former.

The most obvious explanation would be lower incubation temperature (or possibly nutrition?) which is known to slow the rate of development, at least if it's not background genotype (which I'm assuming it isn't, as I assume the authors are saying all genotypes are now "delayed"). If it is a slower rate of development, then the rate of development needs recalibrating, not the reference time.

Given the prior literature (much of it from the Eaton lab) uses a linear time scheme with 0h at the start of pupal life, I'm unsure why the authors didn't just stick with this scheme and rescale the timings so peak elongation is at 23h? This would make the data much more comparable both internally in the manuscript and to past work.

*Reviewer #3 (Recommendations for the authors):*

We believe the manuscript is likely ultimately suitable for publication in *eLife*, but there are a few major comments we would like the authors to address before publication.

1. We find the second paragraph of the discussion confusing. In it, the authors highlight that the difference in the initial recoil velocity between wt and core PCP mutants does not lead to a phenotype in tissue morphogenesis. Then they try to explain why this difference does not generate a global phenotype, but the explanation, "the proportionality factor can depend on the genotype and can change in time", is quite unclear.

In the paragraph that starts with line 153, it seems the authors are saying their data suggests that both $k_f$ and $\eta_f$ are changing in the pk30 mutant. Is that the "proportionality factor" they mean above? If so, why not just come out and say more explicitly that the in the mutants the fast viscosity apparently changes along with $k_f$ so as to make the time scale constant? (As a side note isn't that an interesting coincidence? Would it be interesting/useful to speculate that there might be some sort of feedback loop or mechanical coupling that drives the compensating change in the viscosity?)

And we believe the "change in time" clause above refers to the paragraph that starts at line 179, which is also confusing (see next comment). But isn't the change in viscosity alone already enough to explain the lack of large-scale phenotype? Do you need this change in time also? Would such a change in time be alone sufficient to explain the lack of a global phenotype?

2. The paragraph starting at line 179 is apparently asking the reader to compare the data for the pk30 mutants in Figure 3B (blue data) to the data for the pk30 mutant in S3.1C (blue line). But it's very difficult to do that, as the axes for the two plots are different, and there are not even units given for the y-axis in Figure 3B. We think we're supposed to see that the blue curves in the two plots have different shapes in time. But that's really difficult to see toggling between them. Would it be possible to show both sets of data on the same plot (perhaps one set rescaled) so that we could directly compare the shapes? Moreover, it would be useful to state something more about the data than "its different". It looks like maybe the ESCA data peaks earlier in time than the initial recoil velocity. What does that mean?

3. What is the possible reason for different times for the peaks in anisotropic stress and cell elongation (discussion in the last paragraph of section 2.3)? It seems the response time for the stresses is almost 2 to 4 h.

4. The discussion on ESCA analysis is incomplete. It is not clear how σ/Ks is related to anisotropic ((σ¯/2K)) and isotropic stress (σ/K¯). The variables K,K¯ and σ¯ are not defined. It is mentioned in the caption of Figure 2 that the equations for anisotropic and isotropic stress are on the right side. However, there is no equation in Figure 2F.

---

## [Author Response]

Essential revisions:1) Please revise the text to be more clear about how you reconcile your observations that PCP mutants have an effect on the local tissue dynamics (e.g., recoil velocity upon laser ablation) but this has no global consequences on wing morphogenesis. In particular, could the local effects explain the subtle, but significant effects on adult wing shapes in these mutants?

We now mention this possibility in the Results section, lines 102-110, and then again in the discussion, 242-248. In short, our result that PCP mutations only affect k_f is consistent with the lack of strong phenotype at the tissue scale. Whether changes in k_f can lead to more subtle defects in tissue morphogenesis is possible, but we have no mechanism as yet to propose, and this would require a much larger sample size to address.

2) Please justify your choice of pk30 mutants for most of the analysis, instead of using stbm6 or fmi mutants that completely disturb the core-PCP pathway.

Most of the analysis was actually done with all three mutants: analysis of cell dynamics (Figure 1, Figure 1-figure supplement 2D, Figure 1-figure supplement 3, Figure 1-figure supplement 4), tissue shape (Figure 1-figure supplement 2C,E, Figure 1-figure supplement 4H), adult wing shape (Figure 1-figure supplement 5I, J), initial recoil velocity over time (Figure 2B, Figure 2-figure supplement 1A, Figure 3A, Figure 3-figure supplement 1A), cell elongation over time (Figure 3D, Figure 3-figure supplement 2E). The only analysis that was done with the *pk* mutant alone was the full dynamics of the response to laser ablation and the comparison of initial recoil velocity over time with ESCA (Figure 2C-F’’, Figure 2-figure supplement 1BC, Figure 3-figure supplement 1B-D). This mutant was chosen because the line grows the most reliably in our hands, but since none of the other results differed between mutants, we have no reason to expect any differences in these more focused analyses or in our conclusions about the core PCP pathway in general.

3) Please explain in your rheological model why you believe that the recoil velocity has two phases.

We have added an explanation in the Results section (lines 129-34), as well as a new supplemental figure (Figure 2—figure supplement 1B) to address this point. We also provide a detailed response to this point in answer to Reviewer 2 point 13.

4) Please reconsider the way you plot your data so that readers do not need to extract subtle differences on curves plotted in different graphs with different scales.

This is useful feedback, and we have changed the most relevant figures (Figure 3—figure supplement 1C and Appendix Figure 1).

4) Please revise the discussion on the ESCA analysis, as reviewers have a hard time following it.

Thank you for pointing out the lack of clarity. We have considerably changed the text of the results, lines 161-75 and 199-212, to amend this shortcoming.

Reviewer #1 (Recommendations for the authors):Overall the manuscript is well written and the results support the authors' conclusions; therefore I am supportive of publication.

We thank the reviewer for a careful reading of the manuscript and for helpful and critical feedback.

Here, I list a few comments that will improve the manuscript's readability, but will likely not affect the conclusions of the study:– Considering that pk30 mutants maintain the polarity towards the margin i.e., both core-PCP and Fat-PCP remain aligned (Figure S1.1), while stbm6 and fmi mutants completely disturb the core-PCP pathway, why did the authors focus their analysis in the pk30 mutants (line 118)? If the purpose of the study is to examine the perturbations of the core-PCP, why not choose stbm6 or fmi mutants? The authors should justify this further.

To clarify, most of the analysis (shown in Figure 1 with Figure 1—figure supplements 2-5, Figure 2B with Figure 2—figure supplement 1A, Figure 3A with Figure 3—figure supplement 1A, and Figure 3D with Figure 3—figure supplement 2E) is actually performed on all 3 mutants (pk, stbm, fmi), and they behave similarly. We find the effect on overall cellular dynamics, tissue shear, as well as initial recoil velocity over developmental time are all the same. The only place in which we analyze the pk mutant alone is the full dynamic analysis of the response to laser ablation used for the model (Figure 2C-E’’’) and the comparison of ESCA with initial recoil velocity (Figure 2F-F’’, Figure 2—figure supplement 1B-C, Figure 3—figure supplement 1B-D). We altered the title of section 2.3 to reflect this fact (previously it only referenced wt and pk mutants, whereas now we say wt and core PCP mutants). We also slightly altered the paragraph structure so that now all the Pk-only data in 2.3 are in one paragraph (lines 199-212). Pk mutant flies were best suited for this experiment because they grow more reliably and therefore we could obtain data with better statistics.

– While it is clear that this study only focuses on the core-PCP pathway, perturbing only this pathway will not eliminate polarity and therefore it is not so surprising that oriented cell rearrangements persist. The authors should discuss which experiments (mutants) they suggest to perturb polarity altogether.

We agree with the reviewer that we are not completely eliminating polarity, as there may be many other sources of polarity. We can only conclude from our data that core PCP is not required to orient cell dynamics. We mention two likely additional sources of planar polarity already in the manuscript, including Fat PCP and Toll-like receptors (lines 255-58), which may act independently or redundantly with core PCP or each other. Perturbing all sources of polarity would be very difficult and beyond the scope of this manuscript, as it would involve complicated genetics to combine multiple different types of mutations, which may not even be viable. In addition, it is still possible that there are unknown planar polarity pathways. Furthermore, anisotropic mechanical stress induced by hinge contraction could itself provide a source of polarity through mechanosensitive activity of the cytoskeleton, a possibility that we now suggest in the Discussion on lines 262-7.

– The authors chose a specific region of the wing (between veins L3 and L4) to perform their perturbations. The justification is that this is a region that is easy to identify. But, do the results depend on this choice? Isn't the strength of the polarity signal larger closer to the wing margin?

We cannot rule out the possibility that the results depend on the choice of location, since we did not do different locations. However, in contrast to what the reviewer suggests, the strength of core PCP polarity actually does not seem to vary significantly in space and there is no evidence of stronger polarity signal near the wing margin, according to data in Merkel et al. 2014 (Figure 1D, G, J – quantification of polarity nematics across the wing over time in Stbm::eYFP).

– Use consistent English spelling throughout the manuscript (e.g., behaviour or behavior lines 169 vs. 171).

We thank the reviewer for this comment. We have revised and now use US spelling convention.

Reviewer #2 (Recommendations for the authors):The authors have done a good job of trying to bring together what appear to be datasets generated over a fairly long period by different authors.I do have a few issues regarding the presentation and interpretation of the data (specific points noted below). The main concern is that the authors keep flipping between saying core PCP has no effect on wing morphogenesis, to saying it has subtle effects to saying it has effects but they are not statistically significant. This is hard to reconcile with the convincing effect on tissue mechanics reveals by the laser ablation work, and possibly also the changes in adult wing shape and known roles of core PCP in both the wing and other tissues in the fly. Mostly I think it's just a question of careful choice of words and most likely leaving the question of effects of core PCP on morphogenesis open until all the data is considered together in the Discussion e.g. try to stick to observations in the Results and try to only do interpretation in the Discussion.I also have queries about the rheological model and why the authors believe the recoil has 2 phases based on the data they show, and also about the way developmental timings have been recalibrated.

We thank the reviewer for careful reading and well-informed, thoughtful feedback of our work.

Specific points:The references on lines 21 and 22 are an eclectic selection. The sentence seems to be about how polarized PCP proteins orient structures in epithelia, but the references include a mix of reviews and papers, some from before the polarization had been reported. For instance, Eaton 1997 was a seminal review that suggested planar polarization of stereocilia might be under the control of planar polarity pathways, but does not report such a result (I think the first papers were Montcouquiol et al. 2003 and Curtin et al. 2003?). It would probably be better for readers if the authors either cited reviews or if they want to cite primary papers, than just the first reports of polarized localization determining the orientation of structures (e.g. I think Usui et al. 1999 is the first in flies? Then Axelrod and Strutt?).

The sentence has been changed and now reads:

“Tissue-scale alignment of this pathway is known to orient cellular structures, such as hairs and cilia, and influence dynamic cellular behaviors during morphogenesis, such as cellular movements and cell divisions, through interactions with the cytoskeleton”.

We now only cite reviews.

The references on lines 27 and 28 are mostly referring to Fat-Dachsous while the rest of the paragraph is about "core". There are papers about core PCP in flies affecting cell rearrangements/divisions. Segalen et al. 2010 is a good example, but not the first to report core PCP controlling SOP division orientation (Gho and Schweisguth 1998?). Sugimura and Ishihara 2013 report changes in cell rearrangements in the wing of a core PCP mutant. Others (moving away from the wing/notum) might include many papers about ommatidial rotation in the eye, joint formation in the legs, cell rearrangements in the tracheal system (and probably others). A better review of core PCP roles in tissue morphogenesis in flies would put the work in a better context.

We thank the reviewer for this feedback. We have now changed the introduction, shortening that first paragraph to the sentence mentioned in point 1, and removing references to the Fat/Ds pathway. We have also added a more focused introduction to core PCP’s role in morphogenesis in *Drosophila* on lines 50-59, including many of the references you suggest.

In general, I would recommend thinking carefully about the references in the Introduction so that they are of the most help to readers from outside the field. It is hard to see an organizing principle behind the mixture of reviews from different decades, primary papers (but often not first reports), cloning, or genetic or biochemical characterization studies. As above, I would recommend recent reviews (where available) and primary papers that are the first reports, for each point the authors wish to make.

We have revisited the references and adhered to the advice of the reviewer, focusing on reviews or first reports.

Line 44 – Fz is not a cadherin.

Fixed.

Figure S1A -I don't fully agree that 30 hAPF is the "End of morphogenesis" as suggested by the figure label. Even if you ignore the emergence of hairs and wing ridges, the epithelium itself will go on expanding and folding in subsequent hours. Similar statements are made on line 73 and on Figure 1.

We have now introduced the term 'end of blade elongation flows' to describe the end of our experimental time-window and have adjusted the text and figures accordingly.

Figure S1A legend – do "tissue flows reorganize Fat PCP"? My guess is Fat PCP continues to follow the Ds and Fj gradients and its insensitivity to the tissue flows is why it ends up perpendicular to core PCP. Of course, tissue flow might also have an effect but do the authors have evidence for this (I'm not sure Merkel et al. show this?). The statement in the legend is also not the same as what lines 51-52 say, where the message is that Fat PCP remains margin oriented.

We thank the reviewer for noticing this error, which we have now corrected.

Lines 66 and 67 – in normal nomenclature pk30 = pk[30] (i.e. 30 in superscript) and stbm6 = stbm[6], consistent with fmi[frz3]. I think it would be simplest to refer to stbm[6] as stbm and pk[30] as pk after the first definition, in the same way, fmi[frz3] is referred to as fmi, but this is of course up to the authors. As an aside, fmi[frz3] was described as a strong hypomorph in Chae et al. 1999, so it seems unlikely that in frz3/frz3 the "PCP network is absent", as stronger phenotypes are seen for frz3/Df, implying some residual activity.

This is a good suggestion. We followed the advice of the reviewer and altered the nomenclature in text and figures as suggested. We also now refer to *fmi* and *stbm* as strong hypomorphs that greatly/strongly reduce core PCP.

Figure S1.2C – typo in the first word of the label.

Fixed.

Line 84 "these subtle changes are not statistically significant". This all looks reasonable, but possibly the authors need to qualify this by saying something like "considering the small numbers of wings analyzed". I appreciate the work involved to get these datasets was enormous, so higher n numbers are not feasible. However, with larger n numbers there might be a difference. The average shear values for all three mutant conditions are lower than wt. Ideally, a power calculation would be done to place a numerical value on the likelihood of the shear being the same in each condition, but again I appreciate this is difficult and possibly not appropriate post hoc.

We have revised the text (lines 98-111) and included a new subsampling of the adult wing data to estimate the likelihood of detecting a significant result in pupal stages with small N (Figure S1.5J). If PCP does indeed have a subtle effect on pupal morphogenesis, by the same magnitude as is visible in the adult wing shape, we estimate that we would detect such a difference only ~20% of the time with N=3. Thus, as the reviewer suggests, we have limited statistical power with only N=3. Therefore, while we can rule out the possibility that PCP is absolutely required for the global pattern of cell dynamics and tissue shear, as these are largely unchanged, we cannot rule out the possibility that PCP has subtle effects at this stage.

Line 85 "also do not show differences" – see the previous point. Without stats, you can't say they don't differ, but here it is implied that the data in Figure S1.2C and D have been tested for differences and none was found. This is true for S1.2C with the caveat of small n numbers and no power calculation, but for S1.2D appears to be a judgment made by eye. However, by eye total tissue area at -2h looks lower for pk[3] and stbm[6] than for wt, and at e.g. 6h all 3 mutants show lower cell area changes than wild-type. I think the most the authors can say (unless I've missed something) is that the patterns of tissue behavior look broadly the same for the mutants as for wt, suggesting no gross differences. Note lines 89-90 say "Again, we find only subtle differences" which is not the same as no differences.

We now include a statistical test for the accumulated isotropic deformation by the end of the experiment (Figure 1—figure supplement 2E). There are no significant changes. We have changed the text as suggested to say the patterns are “broadly the same” (line 92). We have also shortened the discussion of the regional analysis along the PD axis (previous lines 89-90, now lines 93-97), removing the part about subtle differences.

Lines 94-95 and Figure S1.4G – again I think it is deceptive to put weight on statistics with low n numbers and no power calculation. I'm not confident that you can "conclude that core PCP does not guide the global patterns of cell dynamics during pupal morphogenesis". The best that you can conclude is that changes are at best subtle (and in the experiments presented might be due to "noise" but could also be due to roles for core PCP in cell behavior). The authors rather undermine themselves by immediately going on to say that there is a reproducible difference in adult wing shape, although admittedly we don't know that this is due to events between 16-32h as the authors note. I think more nuanced conclusions would be better.

We respond below together with point 12.

Lines 115-116 "cellular dynamics is [are?] basically unperturbed" is another example of wording that implies the authors think cellular dynamics are perturbed to some extent, but they want to minimize the significance of this observation. Similarly, lines 198-199 "no significant phenotype" belies the difficulty in making such an assertion when the genotypes do look different in some aspects and the tests have (apparently) minimal statistical power.

Given that we do not see considerable changes in the cellular dynamics in core PCP mutants, we are confident in concluding that core PCP is not the sole determinant of these patterns. We agree, however, that it remains possible that there are more subtle effects of PCP mutation on cell flows at this stage, which would only be confirmed with better statistics. We now state this point in the text (lines 102-11, 229-30, 242-48). However, this fact does not have an impact on the main results of our work, namely, that core PCP is not required for organizing large scale tissue flows.

Lines 119-120 "When plotting displacement of the nearest bond to ablation over time, we observe both a fast (<1 s) and a slow (<20 s) regime (Figure 2C)." I think the authors are saying that their data fits a 2 phase exponential with a fast and slow phase, but I'm a bit nervous about this as the first phase seems to be based on a single data point at 1 sec. I'm not an expert on curve fitting, but only having 1 point to demonstrate the presence of a "fast" phase seems a bit unsafe. I couldn't find anything in the manuscript that describes why the authors decided a 2 phase fit was the correct choice. The subsequent model appears to assume 2 phases, but I'm interested to know if you can get just as good a fit with 1 phase and if you build a model with 1 phase does that still show pk30 having a significantly lower displacement over time? I'm concerned that a 1 phase model may have been unnecessarily ruled out.

Thank you to the reviewer for pointing out this oversimplification in our previous version. We now include a new supplemental figure (Figure 2—figure supplement 1B), which better demonstrates the need for a double exponential, as well as an explanation in the Results section (lines 129-34). In short, the single exponential fit is not sufficient (as shown in Figure 2—figure supplement 1B, right). The fast timescale (<1~s) is required to account for first 5-10 datapoints, not just the first.

Appendix 1 – I understand the issue confronting the authors. Earlier studies from the lab (from Aigouy et al. 2010 onwards) consistently saw peak cell elongation at ~23h, then in Iyer et al. peak elongation appears to be at ~28h (I'm unsure if Iyer et al. comments on this). I have a couple of comments:Line 519 the statement "we identified a delay on the onset of pupal wing morphogenesis" is unclear to me. Do the authors mean that morphogenesis proceeds at a normal rate but just start late (similar to a train being delayed but running at the normal speed), or is everything just happening slower? The curves in panels A and B really look like the latter (e.g. in panel B the same sequence of events in 2016 takes 12h while in 2018 it takes 20h), but the authors seem to be saying it's the former.

Yes, by saying that there is a delay in the onset of the process, we mean that the train is delayed but running at the same speed. Note that we have altered this figure to now include two more 2018 videos (fmi genotype), in response to comments from Reviewer 3. This means that we can now compare 3 (1 *wt*, 2 *fmi*) 2018-2019 videos to the average of our 2016 *wt* videos. To better facilitate comparison, we have now added landmark gridlines at particular locations: the start and end of the 2016 videos, as well as the crossing of the curves for cell rearrangements and cell elongation changes.

Note that the reviewer’s estimation of total duration of the process – 12h in 2016 (from -4 to 8hr) vs 20h in 2018 (from -12 to 8hr) – assumes that the process has only begun at the beginning of the experiment. This is true for the later videos, but not for the 2016 videos, where the rate of shear is already high at the start of the video. Thus, we cannot know from the 2016 videos how long the actual process takes, as we do not see the start point. It is better to compare the duration of processes that are visible in both sets of videos. For example, the duration of time spanning from the point where the curves for the cell elongation changes and cell rearrangements cross each other (dashed to second dotted line) is approximately 9hr in both 2016 and 2018 videos. Thus, we conclude that the delay in the onset of the process is a far greater difference than the difference in overall rate of the process.

We have performed many experiments to try to sort out the cause of the delay, which we recently published as a preprint. We now cite this work in the Appendix.

The most obvious explanation would be lower incubation temperature (or possibly nutrition?) which is known to slow the rate of development, at least if it's not background genotype (which I'm assuming it isn't, as I assume the authors are saying all genotypes are now "delayed"). If it is a slower rate of development, then the rate of development needs recalibrating, not the reference time.

We have actually done quite some experiments to test such hypotheses. We found that the significant shift in the peak of cell elongation between 2016 and 2018 cannot be attributed to a difference in temperature (either on the microscope or during development), nutrition (plant vs yeast-based foods), genetic background (now shown with the *fmi* videos in the updated Appendix figure), presence of *Wolbachia*, or circadian gating. With all these perturbations, we see more cell elongation at 28hr than at 24hr, in contrast to our 2016 data. Our best guess is that the change in developmental timing results from the change in the wavelength of light emitted by the incubators. In fact, two incubators at the same temperature/humidity with different light emission spectra produce slightly different timecourses of cell elongation, although this behavior has not yet been fully characterized. As these experiments are tangential to the topic of this paper, we did not include them. Some of the authors have now prepared a manuscript containing these data and have published it as a preprint, which we now cite in this revised manuscript. As noted above, the delay is more prominent than the change in rate, justifying our choice to merely shift the curves, rather than rescale.

Given the prior literature (much of it from the Eaton lab) uses a linear time scheme with 0h at the start of pupal life, I'm unsure why the authors didn't just stick with this scheme and rescale the timings so peak elongation is at 23h? This would make the data much more comparable both internally in the manuscript and to past work.

Indeed, this is what we have done in the past – all timelapse data were aligned on the peak of cell elongation to make them more comparable and account for slight differences in staging, but we called them all 16hr APF. In the past, however, the shift was only ~30min and thus referring to them all as “16hr APF” was still reasonable. Now, however, if we apply the same strategy, we would have to shift by ~6-8hrs. Continuing to call the peak of cell elongation “16hr APF” at that point, when in some videos that was actually 23hr APF, would likely cause considerable confusion and makes the “hAPF – after puparium formation” reference rather meaningless. Our data from Appendix 1 and the new preprint indicate that there is something happening after the onset of pupariation that delays the onset of pupal flows, but that once the flows start, they occur at more-or-less the same rate. Since the onset of pupal flows is not visible in all of the videos (in particular the 2016 data, when at the time we start imaging the shear rates are already high), we chose to orient the data based on the peak of cell elongation. We now introduce a relative time scale called “hRPCE” for hr relative to peak cell elongation.

Reviewer #3 (Recommendations for the authors):We believe the manuscript is likely ultimately suitable for publication in eLife, but there are a few major comments we would like the authors to address before publication.

Thank you for the careful reading of the manuscript and the excellent suggestions for improvement.

1. We find the second paragraph of the discussion confusing. In it, the authors highlight that the difference in the initial recoil velocity between wt and core PCP mutants does not lead to a phenotype in tissue morphogenesis. Then they try to explain why this difference does not generate a global phenotype, but the explanation, "the proportionality factor can depend on the genotype and can change in time", is quite unclear.

We have now revised the discussion, as well as the results, in response to this and later comments. In the discussion, we now first address the phenotype on initial recoil velocity and why it may not be reflected in global tissue stress or cell dynamics. We have taken out the confusing statement about proportionality between stress and recoil velocity. Instead, we merely point out that recoil velocity is often used as a proxy for stress in other works, and that here we highlight a limitation of that approach in looking at differences between genotypes, as recoil velocity can also report differences in fast timescale mechanics that are actually not reflected in tissue stress.

In the paragraph that starts with line 153, it seems the authors are saying their data suggests that both $k_f$ and $\eta_f$ are changing in the pk30 mutant. Is that the "proportionality factor" they mean above? If so, why not just come out and say more explicitly that the in the mutants the fast viscosity apparently changes along with $k_f$ so as to make the time scale constant? (As a side note isn't that an interesting coincidence? Would it be interesting/useful to speculate that there might be some sort of feedback loop or mechanical coupling that drives the compensating change in the viscosity?)

No, this is not the proportionality factor we meant above. The proportionality factor was meant to describe how tissue stress may be proportional to recoil velocity. The reviewer is correct, however, that k_f changes with eta_f. We have now clarified this point, in particular we speculate that this proportionality suggests that both parameters originate from a single microscopic mechanism that constrains the relaxation time eta_f/k_f, such as cytoskeletal turnover time (lines 174-80).

And we believe the "change in time" clause above refers to the paragraph that starts at line 179, which is also confusing (see next comment). But isn't the change in viscosity alone already enough to explain the lack of large-scale phenotype? Do you need this change in time also? Would such a change in time be alone sufficient to explain the lack of a global phenotype?

We agree with the referee that this phrase was confusing and we have removed it.

2. The paragraph starting at line 179 is apparently asking the reader to compare the data for the pk30 mutants in Figure 3B (blue data) to the data for the pk30 mutant in S3.1C (blue line). But it's very difficult to do that, as the axes for the two plots are different, and there are not even units given for the y-axis in Figure 3B. We think we're supposed to see that the blue curves in the two plots have different shapes in time. But that's really difficult to see toggling between them. Would it be possible to show both sets of data on the same plot (perhaps one set rescaled) so that we could directly compare the shapes? Moreover, it would be useful to state something more about the data than "its different". It looks like maybe the ESCA data peaks earlier in time than the initial recoil velocity. What does that mean?

We thank the referee for the good suggestion, and we have now updated the supplemental figure (Figure 3—figure supplement 1D) to plot the relevant data in similar styles, side-by-side, to facilitate a direct comparison. We have now re-written the results to better present the ESCA method as well as the comparison between the ESCA and the initial recoil velocity results (lines 161-76, 199-212).

3. What is the possible reason for different times for the peaks in anisotropic stress and cell elongation (discussion in the last paragraph of section 2.3)? It seems the response time for the stresses is almost 2 to 4 h.

In previous work (Etournay et al., *eLife* 2015), we measured the relationship between initial recoil velocity and cell elongation, and we found that they follow a linear relationship with an offset. We have interpreted this offset as a signature of active stress in the tissue, but at the time we treated it as a constant in time. However, current results highlight the possibility that the active stress is dynamic. This opens interesting questions for future work, in particular the possibility that active stresses in the pupal wing are mechanosensitive, as we have recently proposed to account for cell elongation patterns in the larval wing disc (Dye et al., *eLife* 2021). We now explain this in results (lines 216-220) and in the discussion (lines 262-67).

4. The discussion on ESCA analysis is incomplete. It is not clear how σ/Ks is related to anisotropic ((σ¯/2K)) and isotropic stress (σ/K¯). The variables K, K¯ and σ¯ are not defined. It is mentioned in the caption of Figure 2 that the equations for anisotropic and isotropic stress are on the right side. However, there is no equation in Figure 2F.

We thank the reviewer for pointing out this confusion. We have rewritten this section of the Results, including a more extensive description of ESCA and why we use it (lines 161-76). We have also adjusted the variable names to prevent confusion between the parameters in ESCA and our rheological model and now introduce them in the text.